# A new family of bacterial ribosome hibernation factors

Karla Helena-Bueno[1,12], Mariia Yu. Rybak[2,12], Chinenye L. Ekemezie[1], Rudi Sullivan[3], Charlotte R. Brown[1], Charlotte Dingwall[1], Arnaud Baslé[1], Claudia Schneider[1], James P. R. Connolly[1], James N. Blaza[4,5,6], Bálint Csörgő[7], Patrick J. Moynihan[3], Matthieu G. Gagnon[2,8,9,10 ✉], Chris H. Hill[5,6,11 ✉] & Sergey V. Melnikov[1 ✉]

To conserve energy during starvation and stress, many organisms use hibernation factor proteins to inhibit protein synthesis and protect their ribosomes from damage[1,2]. In bacteria, two families of hibernation factors have been described, but the low conservation of these proteins and the huge diversity of species, habitats and environmental stressors have confounded their discovery[3–6]. Here, by combining cryogenic electron microscopy, genetics and biochemistry, we identify Balon, a new hibernation factor in the cold-adapted bacterium *Psychrobacter urativorans*. We show that Balon is a distant homologue of the archaeo-eukaryotic translation factor aeRF1 and is found in 20% of representative bacteria. During cold shock or stationary phase, Balon occupies the ribosomal A site in both vacant and actively translating ribosomes in complex with EF-Tu, highlighting an unexpected role for EF-Tu in the cellular stress response. Unlike typical A-site substrates, Balon binds to ribosomes in an mRNA-independent manner, initiating a new mode of ribosome hibernation that can commence while ribosomes are still engaged in protein synthesis. Our work suggests that Balon–EF-Tu-regulated ribosome hibernation is a ubiquitous bacterial stress-response mechanism, and we demonstrate that putative Balon homologues in *Mycobacteria* bind to ribosomes in a similar fashion. This finding calls for a revision of the current model of ribosome hibernation inferred from common model organisms and holds numerous implications for how we understand and study ribosome hibernation.

When a living cell encounters environmental stress, its metabolic activity is greatly reduced until conditions improve. Until recently, this was believed to be a passive process in which enzymes simply become idle, with vacant active sites. It is now clear that organisms across the three domains of life use specific mechanisms to inactivate or protect critical cellular machinery from damage during stress. This involves placing key enzymes into a controlled state of molecular hibernation[1–9].

Most extensively, the process of molecular hibernation has been studied in ribosomes—essential ribonucleoprotein complexes that catalyse protein synthesis, also known as translation. During normal conditions, ribosomes bind their ligands, such as mRNA and tRNAs, to carry out protein synthesis[10]. However, during starvation and stress, ribosomes dissociate from mRNA and tRNAs and enter molecular hibernation by associating with proteins known as hibernation factors[1,2]. Ribosome hibernation factors prevent protein synthesis by occupying ribosomal binding sites for mRNA and tRNAs and protect ribosomes from degradation by shielding their vulnerable active centres from cleavage by cellular nucleases[5,6,11,12]. Hibernation factors thereby allow cells to rapidly switch between the states of active growth and dormancy[13–15].

Although several families of ribosome hibernation factors have been described so far, it is unknown how many exist in nature. Eukaryotic examples include proteins Stm1 (also known as Serbp1), Ifrd2, Lso2 and Mdf1 present in various lineages of mammals and fungi[16–20]. Two main families of hibernation factors exist in bacteria, including the RaiA family, common to many bacterial lineages, and RMF, found in some γ-proteobacteria[2]. Unlike most core translation factors, hibernation factors are highly diverse and structurally dissimilar, lacking conservation even within a single domain of life[6,17–21]. This has complicated the discovery of new hibernation factors, and an over-reliance on common model organisms—mainly *Escherichia coli*—leaves it unclear to what extent current findings can be considered generally representative.

To address this, here we introduce a new model organism: the cold-adapted bacterium *P. urativorans*. By examining *P. urativorans* ribosomes isolated under conditions of cold shock and stationary phase, we discover a new mechanism of translational response to

[1]Biosciences Institute, Newcastle University, Newcastle upon Tyne, UK. [2]Department of Microbiology & Immunology, University of Texas Medical Branch, Galveston, TX, USA. [3]School of Biosciences, University of Birmingham, Birmingham, UK. [4]Department of Chemistry, University of York, York, UK. [5]York Structural Biology Laboratory, University of York, York, UK. [6]York Biomedical Research Institute, University of York, York, UK. [7]Synthetic and Systems Biology Unit, Institute of Biochemistry, HUN-REN Biological Research Centre, Szeged, Hungary. [8]Department of Biochemistry & Molecular Biology, University of Texas Medical Branch, Galveston, TX, USA. [9]Sealy Center for Structural Biology & Molecular Biophysics, University of Texas Medical Branch, Galveston, TX, USA. [10]Institute for Human Infections & Immunity, University of Texas Medical Branch, Galveston, TX, USA. [11]Department of Biology, University of York, York, UK. [12]These authors contributed equally: Karla Helena-Bueno, Mariia Yu. Rybak. ✉e-mail: magagnon@utmb.edu; chris.hill@york.ac.uk; sergey.melnikov@newcastle.ac.uk

stress. Using cryogenic electron microscopy (cryo-EM), we observe that *P. urativorans* ribosomes bind to an uncharacterized protein that occupies their active centres. Strikingly, in contrast to previously identified hibernation factors, this factor can engage not only with vacant ribosomes but also with ribosomes associated with mRNA and tRNAs.

Homologues of this new factor are present in nearly 20% of studied bacteria, although notably absent from common model organisms such as *E. coli* and *Staphylococcus aureus*—explaining why it has been undetected until now and emphasizing the importance of venturing beyond typical mesophilic bacteria to discover new biology. Our cryo-EM analysis of this factor from *P. urativorans*, *Mycobacterium smegmatis* and *Mycobacterium tuberculosis* indicates that this protein is a distant homologue of the archaeo-eukaryotic translation factors aeRF1 and Pelota that participate in other aspects of the translation process, not hibernation. We therefore name this protein Balon (after balón, Spanish for ball) to highlight its distant structural similarity to Pelota (also Spanish for ball). This discovery of Balon demonstrates that bacteria can use a qualitatively distinct and previously unknown mechanism of translational stress response compared to the generalized model of ribosome hibernation based on studies in *E. coli*, offering broad implications for how we understand and study the process of ribosome hibernation in response to stress.

## A new ribosome hibernation factor, Balon

Our interest in ribosome hibernation arose from our studies of protein synthesis at cold temperatures, using the bacterium *P. urativorans* as a model organism. Members of the *Psychrobacter* genus are notorious organisms that can spoil refrigerated food owing to their ability to grow at sub-zero temperatures[22,23]. In the laboratory, *P. urativorans* has a recommended growth temperature of 10 °C, but in nature it can survive much colder environments, including Antarctic soil[22,23].

To understand how *P. urativorans* adapt protein synthesis to sudden changes in temperature, we isolated their ribosomes shortly after inducing cold shock. As *P. urativorans* are slow-growing bacteria, with a doubling time that can exceed 1 day, we first grew cultures for 7 days at 10 °C to produce sufficient biomass. We then induced cold shock by exposing these cultures to ice for 30 min before extracting ribosomes for proteomic and cryo-EM analyses (Methods and Extended Data Fig. 1). Subjecting *P. urativorans* cultures to ice treatment resulted in a rapid depletion of polysomes and accumulation of monosomes, indicating the inactivation of protein synthesis (Fig. 1a). Our cryo-EM analysis revealed that the ribosomal A site was occupied by a previously uncharacterized protein (Fig. 1b and Extended Data Figs. 2–4). Mass spectrometry identified this as the 41-kDa protein AOC03_06830 (supplementary datasets 1 and 2; supplementary datasets 1–9 are available at https://figshare.com/s/374a95769c5f7e9cdc04), which we rename Balon.

The cryo-EM data, in conjunction with our mass spectrometry analysis (supplementary datasets 1 and 2), showed that about 44% of Balon-containing ribosomes were also bound to EF-Tu, the universally conserved translation factor known to deliver aminoacyl-tRNA molecules to the ribosomal A site during protein synthesis (Fig. 1b and Supplementary Fig. 1).

For the Balon-bound ribosomes, particle classification revealed that nearly two-thirds were also associated with the ribosome hibernation factor RaiA (state 1), whereas another third did not contain RaiA but were instead bound to mRNA and tRNA (state 2; Fig. 1b). The tRNA-bound ribosomes also contained density in the nascent peptide tunnel, indicating the presence of peptidyl-tRNA. In the mRNA channel, density for mRNA could be observed within the A, P and E sites, with the P-site signal showing a well-defined tRNA–mRNA base pairing that corresponds more to heterogeneous rather than a specific type of tRNA or mRNA sequence. Overall, these data suggested that Balon can associate not only with vacant but also with elongating ribosomes.

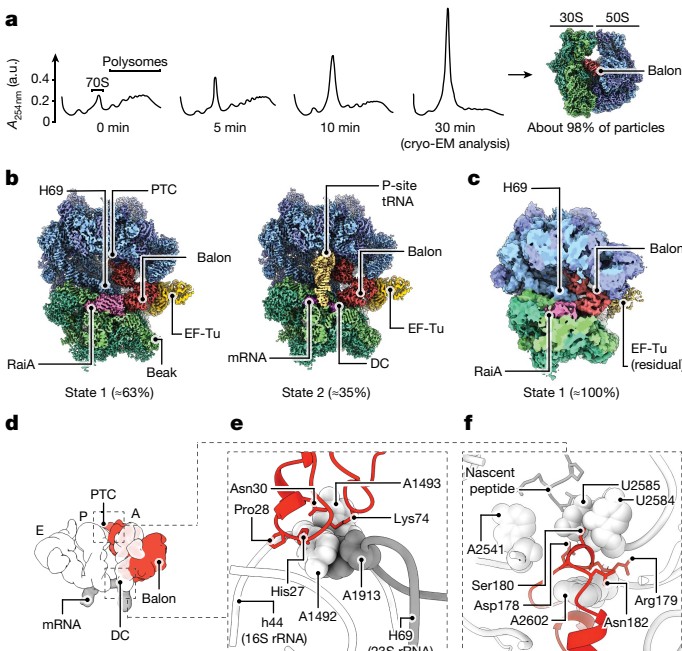

**Fig. 1 | Cryo-EM analysis of ribosomes from cold-adapted bacteria identifies a new ribosome hibernation factor, Balon. a**, Polysome profiling in sucrose gradients shows an accumulation of monomeric ribosomes during the first 30 min of response to ice treatment of *P. urativorans* cells. Absorbance (*A*) was measured at 260 nm in arbitrary units (a.u.). **b**, Cryo-EM maps at 2.6 Å resolution depicting the two most prevalent states of ribosomes isolated from ice-treated bacteria *P. urativorans*. State 1 consists of ribosomes bound to a previously uncharacterized protein, Balon, and the hibernation factor RaiA. State 2 represents ribosomes bound to Balon, mRNA and P-site tRNA. Both states of the ribosome also show the presence of the elongation factor EF-Tu bound to Balon. PTC, peptidyl-transferase centre; DC, decoding centre. **c**, A cryo-EM map at 5 Å resolution depicting the most prevalent state of ribosomes isolated from *P. urativorans* during late stationary phase. **d**–**f**, Structural snapshots illustrating that Balon occupies ribosomal active centres and overlaps with several drug-binding sites. **d**, Superposition of Balon (red), tRNAs (white) and mRNA (grey) to compare ribosomal binding sites (A, P and E) of these molecules. **e**,**f**, Zoomed-in view of the decoding centre (**e**) and peptidyl-transferase centre of the ribosome (**f**), showing details of ribosome recognition by Balon.

To test whether Balon binding to ribosomes may be induced by other stressors, we allowed *P. urativorans* cultures to transition to the stationary phase and remain stationary for 4 days. We then isolated their ribosomes and found that practically all of them lacked P-site tRNA but were bound to Balon, in addition to RaiA and trace amounts of EF-Tu (Fig. 1c and Supplementary Fig. 3). In all of our structures, Balon contacts two active centres of the ribosome: the decoding centre and the peptidyl-transferase centre (Fig. 1d–f). Thus, Balon exhibits two common characteristics of all previously identified ribosome hibernation factors: it binds nearly all cellular ribosomes under stress conditions, and it occupies the ribosomal active sites, rendering them inaccessible to other molecules.

## Balon is a ubiquitous bacterial protein

Our homology search revealed Balon homologues in approximately 20% of representative bacteria, spanning 1,573 out of 8,761 analysed species from 23 out of 27 bacterial phyla (Fig. 2a,b). These species included many model bacteria such as *M. tuberculosis*, *Bacillus subtilis* and *Thermus thermophilus* (supplementary datasets 3–9). However, species such as *E. coli* and *S. aureus*, commonly used for studying ribosome hibernation, lack Balon homologues, which may explain why Balon was not identified in earlier studies.

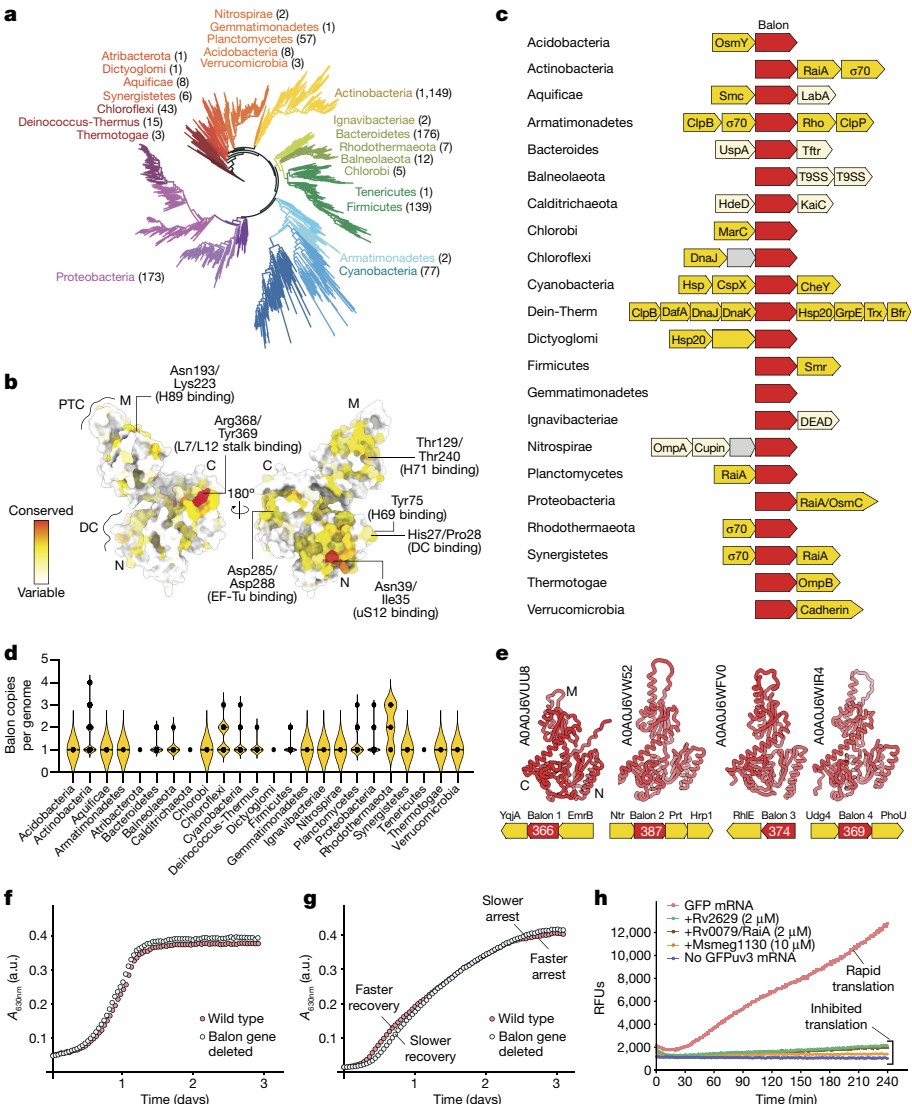

**Fig. 2 | Balon-coding genes are ubiquitous among bacteria and are often found in stress-response operons. a**, The bacterial tree of life shows that Balon homologues are found in most bacterial lineages, encompassing 1,572 representative bacteria from 23 different phyla. **b**, An atomic model of Balon (coloured by sequence conservation) illustrating high conservation of residues responsible for ribosome recognition. **c**, Operon schematics illustrating the genetic context of Balon-coding genes (red) in selected bacterial phyla. Balon-coding genes are typically found in operons that encode stress response factors (light orange). These include chaperones (Hsp20, DnaK and DnaJ), alternative σ70 factors, factors of acid tolerance (HdeD) and osmotic stress tolerance (OsmB and OsmY), ribosome hibernation (RaiA), ribosome repair (RtcB) and multidrug resistance (Smr, MarC and EmrB). Dein-Therm, Deinococcus-Thermus. **d**, Violin plots showing that in 603 species (38% of the analysed species), Balon homologues are encoded by two, three or four gene

copies located in different genomic loci. **e**, As an example of a genome with several Balon orthologues, here we depict the four operons encoding Balon orthologues from *Mycolicibacterium chubuense*. Notably, one of these copies (Balon 1) resides in an operon with the multidrug export protein EmrB, and another copy (Balon 2) is located in an operon with hypoxia-response factors. Their predicted structures (AlphaFold) indicate a common core architecture. **f**, Growth curves of the wild-type and Balon-deficient strains of *P. arcticus* in a rich growth medium. **g**, Growth curves of the wild-type and Balon-deficient strains of *P. arcticus* during their recovery from the long-term stationary phase (3 months). **h**, Plot illustrating translation of the reporter protein GFP in the absence and presence of the hibernation factor RaiA (also known as Rv0079) or mycobacterial homologues of Balon—Rv2629 and Msmeg1130. RFUs, relative fluorescence units.

Although none of these Balon homologues has been functionally characterized, some of them have been annotated as putative stress-response proteins. For example, the Balon homologue YocB in *B. subtilis* is transcriptionally induced by heat shock, cold shock and stationary phase[24]. In *M. tuberculosis* and *M. smegmatis*, the Balon homologues Rv2629 and Msmeg1130 are transcriptionally activated in response to hypoxia, increasing bacterial survival and pathogenicity[25–27].

Balon-coding genes have different genetic surroundings in different phyla but are typically located in stress-response operons (Fig. 2c). Most frequently, Balon-coding genes are located adjacent to the gene for RaiA. Other common genetic neighbours encode response

factors to thermal shock (Hsp20), osmotic stress (OsmC and OsmY), acid stress (HdeD) or antibiotics (MarC and EmrB) or factors of rRNA repair from nucleolytic damage (RtcB)[28,29] (Fig. 2c). We also found that many bacteria (603 species) possess several copies of Balon-coding genes, ranging from 2 to 4 copies per genome (Fig. 2d). For example, some *Mycobacteria* bear up to four copies of Balon-coding genes, with one of them located in the vicinity of the hypoxia-response factor Hrp1, and another being adjacent to the gene for the multidrug transporter EmpB (Fig. 2e).

To verify that some of these Balon homologues are indeed ribosome-binding proteins, we recombinantly expressed the proteins

Rv2629 and Msmeg1130 from *M. tuberculosis* and *M. smegmatis*, respectively. We then determined their structures bound to *M. smegmatis* ribosomes (Supplementary Figs. 3 and 4). We found that both Rv2629 and Msmeg1130 bind the ribosomal A site and share several structural characteristics with Balon that are discussed in detail below, including the absence of aeRF1-like NIKS and GGQ motifs and insertions in the decoding site-binding and the EF-Tu-binding sites (Supplementary Fig. 5). Overall, this analysis shows that Balon is a widespread and structurally conserved bacterial protein frequently contained within stress-response operons.

## Balon affects cellular growth

As the ribosome hibernation factors RaiA, RMF and Stm1 have been shown to enhance cellular stress survival[11,30,31], we next tested whether Balon could have a similar activity. We first engineered a Balon-deficient strain of *P. urativorans*; however, we could not use *P. urativorans* strains for accurate growth measurements owing to their thermal intolerance and incompatibility with standard laboratory equipment. We therefore produced a Balon-deficient strain of *Psychrobacter arcticus*, a close relative of *P. urativorans* that can tolerate temperatures up to 28 °C (Methods). Under optimal growth conditions, the Balon-deficient *P. arcticus* strain grew slightly faster than the wild-type strain (*n* = 8; Fig. 2f and Supplementary Fig. 6), consistent with previous studies showing that a reduced level of expression of the Balon homologue Msmeg1130 accelerates cellular growth of *M. smegmatis*[27]. To better understand the mechanism behind this growth defect, we tested the effects of purified Balon homologues in an in vitro translation assay. Msmeg1130 (*M. smegmatis*) and Rv2629 (*M. tuberculosis*) inhibited protein synthesis as effectively as the known hibernation factor RaiA (Rv0079), causing a 15-fold reduction in the rate of protein synthesis, as assessed by the relative levels of the reporter protein GFP (Fig. 2h).

We next investigated the effects of Balon deletion on recovery from stress. As the advantages conferred by hibernation factors can typically be observed only after extended periods of stress[11], we allowed *P. arcticus* cultures to enter stationary phase and remain dormant for 3 months. We then transferred dormant cells to fresh medium to resume growth. Balon-deficient cells exhibited a visible growth defect during the first day of the experiment but a slower transition back to the stationary phase on the second day of the experiment, compared to the wild-type strain (Fig. 2g and Supplementary Fig. 6b). However, both the wild-type and Balon-deficient cells showed comparable rRNA levels with no visible differences in ribosome degradation, indicating that Balon is unlikely to substantially affect ribosome stability within our experimental time frame (Supplementary Fig. 7). Overall, these data showed that the Balon-coding gene has two opposing effects on cellular fitness: it is slightly deleterious under optimal growth conditions but beneficial during the early stage of cellular recovery from stress.

## Balon resembles aeRF1-type proteins

To gain insight into the possible evolutionary origin of Balon, we used a structure-based homology search to identify distant Balon homologues across the tree of life (Methods). This search revealed that Balon has close structural homologues among eukaryotic and archaeal factors of protein synthesis from the aeRF1 family (Fig. 3a and Supplementary Table 1). This family includes aeRF1, a translation termination factor that binds to the ribosomal A site and terminates protein synthesis in response to mRNA stop codons[32], and Pelota, a ribosome rescue factor that binds to the A site of ribosomes arrested by aberrant mRNAs[33]. Balon shares just 10% sequence identity with both aeRF1 and Pelota, yet these three proteins have the same domain organization: the carboxy-terminal domain that mediates aeRF1 and Pelota interaction

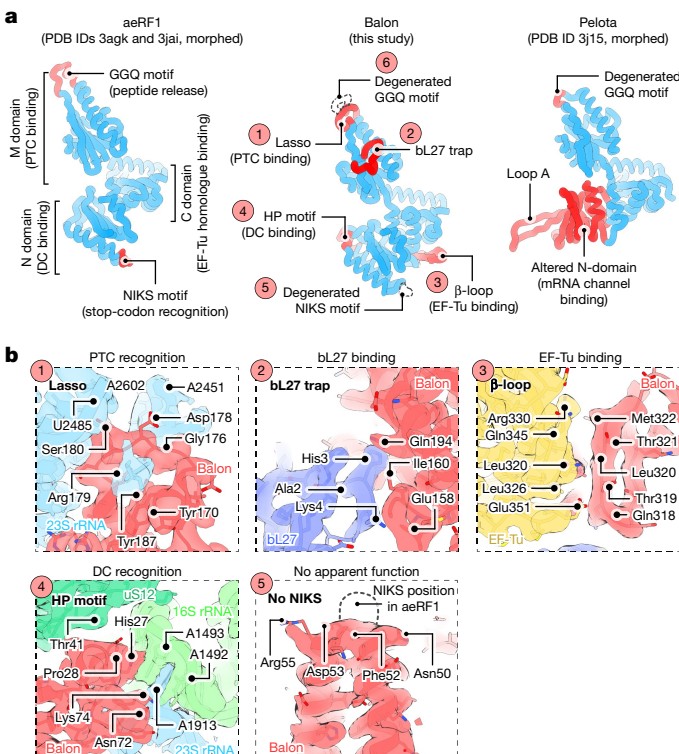

**Fig. 3 | Balon resembles and probably shares its origin with the archaeo-eukaryotic proteins aeRF1 and Pelota. a**, Comparison illustrating structural similarity between Balon, aeRF1 and Pelota proteins. The overall structure of Balon most closely resembles that of the translation termination factor aeRF1: it shares a common domain organization and binds the ribosome in a similar way through contacts with EF-Tu, the decoding centre and the peptidyl-transferase centre. Balon lacks key structural features that are required for aeRF1 activity, including the NIKS motif (stop-codon recognition) and the GGQ motif (nascent peptide release). **b**, Zoomed-in views of the cryo-EM map and the ribosome structure (in state 2) highlighting the molecular contacts between key structural features of Balon and the active centres of the ribosome or EF-Tu.

with EF-Tu and their delivery to the A site; the amino-terminal domain that recognizes the ribosomal decoding centre; and the middle domain that binds to the ribosomal catalytic centre (Fig. 3a,b). This structural similarity suggests a common evolutionary origin for Balon, aeRF1 and Pelota and demonstrates that bacteria possess a hibernation factor from the aeRF1 family.

## Balon binds the decoding centre

To understand how Balon can recognize both vacant and translating ribosomes, we analysed its interactions with the ribosomal A site. Selection of substrates by the ribosome typically depends on the specific sequence of mRNA in the A site. When a stop codon is present, it binds to the characteristic NIKS motif in the N-terminal domain of aeRF1[32] (Fig. 4a). When ribosomes are stalled on truncated mRNAs, the vacant mRNA channel is occupied by the characteristic loop A in Pelota[33] (Fig. 4b). By contrast, Balon lacks both the NIKS motif and the loop A and instead has a distinctive HP motif that directly binds to the decoding centre at residues A1492 in 16S rRNA helix h44, and A1913 in 23S rRNA helix H69 (*E. coli* rRNA numbering; Fig. 4a,b and Supplementary Fig. 8). This binding strategy allows Balon to stay 10 Å away from the mRNA channel, facilitating its unique ability to bind ribosomes independently of whether mRNA is present or not. This also allows Balon to bind ribosomes simultaneously with the

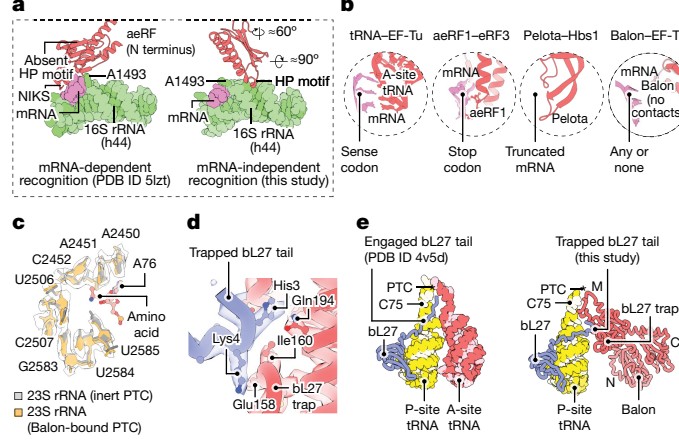

**a**

mRNA-dependent recognition (PDB ID 5lzt)

mRNA-independent recognition (this study)

Absent-HP motif
NIKS
mRNA
A1493
16S rRNA (h44)
aeRF (N terminus)
φ ≈ 60°

A1493
mRNA
θ ≈ 90°
HP motif
16S rRNA (h44)

**b**

tRNA–EF-Tu | aeRF1–eRF3 | Pelota–Hbs1 | Balon–EF-Tu

A-site tRNA
mRNA
Sense codon

mRNA
aeRF1
Stop codon

Pelota
Truncated mRNA

mRNA
Balon (no contacts)
Any or none

**c**

A2450
A2451
C2452
U2506
C2507
G2583
U2584
A76
Amino acid
U2585

□ 23S rRNA (inert PTC)
■ 23S rRNA (Balon-bound PTC)

**d**

Trapped bL27 tail
His3
Gln194
Lys4
Ile160
Glu158
bL27 trap

**e**

Engaged bL27 tail (PDB ID 4v5d)
PTC
C75
bL27
P-site tRNA
A-site tRNA

Trapped bL27 tail (this study)
PTC
C75
bL27
M
bL27 trap
P-site tRNA
N
Balon
C

**Fig. 4 | Balon has a dissimilar mechanism to engage with the ribosomal active sites compared to aeRF1 and Pelota. a,b**, Zoomed-in views illustrating mRNA-independent recognition of the ribosome by Balon. **a**, Comparison illustrating the dissimilar mechanisms by which aeRF1 and Balon (red) recognize the decoding centre of the ribosome. 16S rRNA (green) and mRNA (pink) fragments are shown for simplicity. **b**, Comparison of decoding centre-binding strategies used by Balon and other ligands of the ribosomal A site. **c**–**e**, Zoomed-in views illustrating that Balon-bound ribosomes have an inactive conformation of the catalytic site. **c**, Superposed structures showing that the conformation of the peptidyl-transferase centre in Balon-bound ribosomes is identical to the inert state observed in ribosomes with a vacant A site and a peptidyl-tRNA-bound P site (Protein Data Bank (PDB) code 5mzd; rRNA backbone RMSD of 0.3 Å). **d**, A segment of the cryo-EM map showing bL27 protein (blue) bound to the bL27 trap of Balon (red) in the 70S ribosome (state 2). **e**, Comparison of the ribosomal protein bL27 (blue) in the structure of actively translating ribosomes (left) and ribosomes bound to Balon (right).

hibernation factor RaiA, which would be impossible for aeRF1 or Pelota owing to a steric clash between their N-terminal domains and the RaiA molecule.

## Balon binds the catalytic centre

To understand the impact of Balon on ribosomal catalytic activity, we examined its interactions with the peptidyl-transferase centre. Compared to aeRF1, Balon lacks the characteristic GGQ motif that triggers nascent peptide release from the peptidyl-tRNA[34]. Instead, Balon bears a lasso-like protein loop that wraps around an rRNA nucleotide (A2602 in the 23S rRNA), positioning Balon adjacent to the outer wall of the peptidyl-transferase centre. In the absence of a P-site tRNA (state 1) the lasso loop is poorly ordered, but in the presence of a peptidyl-tRNA (state 2) it is stabilized by contact with the 3′-CCA end of the tRNA. In this position, Balon remains excluded from the catalytic centre by the 23S rRNA residue U2585 (Fig. 4c).

When Balon binds to the A site, the structure of the ribosomal catalytic centre remains inert, as evident from the conformation of 23S rRNA bases U2585 and U2506[35,36] (Fig. 4c). Aside from binding to 23S rRNA residues, Balon also contacts the ribosomal protein bL27, which is absent in eukaryotes and archaea. In bacteria, the N-terminal tail of bL27 binds in the vicinity of the peptidyl-transferase centre and promotes ribosomal catalytic activity by positioning water molecules around ribosomal substrates[36]. However, when Balon binds to the ribosome, it sequesters this N-terminal tail using a unique loop (absent in other aeRF1-type proteins), which we term the bL27 trap (Fig. 4d,e). Therefore, Balon binding to translating ribosomes preserves the catalytic site in its inert state, inaccessible to water molecules. This finding explains the well-defined electron density of the nascent peptide observed in our cryo-EM map (state 2, Extended Data Fig. 4), highlighting Balon's

ability to prevent premature release of nascent peptides by translationally inactive ribosomes.

## Balon and EF-Tu in ribosome hibernation

The observation of EF-Tu bound to *P. urativorans* ribosomes in complex with Balon was unexpected. EF-Tu plays a major role in translation by delivering aminoacyl-tRNAs to the ribosome, and transitions between two conformations to do so. The GTP-bound EF-Tu adopts a closed conformation, facilitating the binding and delivery of aminoacyl-tRNAs to the ribosomal A site. If the aminoacyl-tRNA sequence matches the mRNA sequence, EF-Tu undergoes GTP hydrolysis, transitioning to the open, GDP-bound conformation—thereby releasing the aminoacyl-tRNA into the A site and dissociating from the ribosome[37]. Pelota and aeRF1 are delivered to the ribosome by EF-Tu homologues through a similar mechanism[38].

To investigate the role of EF-Tu in ribosome hibernation, and to gain insights into the mechanism of Balon recruitment to the ribosome, we used focused classification to enrich ribosomes simultaneously bound to both Balon and EF-Tu (about 44% of particles; Extended Data Fig. 2). Domains III and II of EF-Tu are well ordered (Fig. 5a–d and Supplementary Fig. 1). Domain II directly contacts 16S rRNA (helix h5), similarly to previously observed domain II interactions during aminoacyl-tRNA delivery[39] (Fig. 5b). Domain III not only forms interactions with Balon but binds to the tip of the GTPase-activating sarcin–ricin loop, which is a known target for cellular toxins and nucleases[40]. Domain I, containing the nucleotide-binding site, forms previously described interactions with the C-terminal domain of the L7/L12 stalk (consisting of protein bL12)[41] and adopts one of two subtly different conformations in our dataset (Extended Data Fig. 2). Both correspond to the GDP-bound state on the basis of the nucleotide density and the positions of switches I and II (Fig. 5c). The overall conformation of EF-Tu resembles that of the open state of EF-Tu(GDP)[42] (root mean square deviation (RMSD) 1.9 Å) and not the closed state of EF-Tu(GTP)[43] (RMSD 13.8 Å; Fig. 5d). This prompted us to ask whether Balon is delivered to the ribosome through the same mechanism as aminoacyl-tRNA, aeRF1 and Pelota, and if so, why EF-Tu(GDP) rapidly dissociates from aminoacyl-tRNAs, aeRF1 and Pelota, but not from Balon.

To address these questions, we first compared the interaction interface that EF-Tu forms with each of its partners. Whereas Balon, aeRF1 and Pelota all use their conserved C-terminal domain to recognize EF-Tu or its homologues, Balon uses a distinct surface on its C-terminal domain (Fig. 5f). This alternative EF-Tu-recognition site includes a unique β-loop and is about 20 Å away from the corresponding site in aeRF1 and Pelota. Consequently, while in the A site, Balon cannot bind to EF-Tu(GTP), as this would cause a steric clash between the ribosomal sarcin–ricin loop and the closed conformation of EF-Tu(GTP) (Fig. 5e). Furthermore, the stable association between EF-Tu(GTP) and Pelota or aeRF1 requires additional interactions with the middle domain of Pelota or aeRF1 that are possible only with the closed GTP-state of EF-Tu (Fig. 5f). However, the different relative orientation of Balon and EF-Tu(GTP) would preclude the formation of these interactions (Fig. 5f). Without this, the Balon C-terminal domain–EF-Tu interface is limited to about 230 Å², compared to the minimum contact area of about 500 Å² required for a stable interaction[44]. The Balon–EF-Tu complex is therefore unlikely to be stable in solution.

To test our in silico analyses experimentally, we purified EF-Tu and the Balon homologue from *M. smegmatis* (Msmeg1130) and analysed their association in vitro. Unlike other EF-Tu partners, Msmeg1130 could not bind to EF-Tu in the presence of GDPCP (a nonhydrolysable GTP analogue; Fig. 5g and Supplementary Figs. 9 and 10). Furthermore, Msmeg1130 did not form a stable complex with EF-Tu in the presence of GDP (Fig. 5f). We next conducted pelleting assays to assess the ability of Balon homologues and EF-Tu to associate with ribosomes (Supplementary Fig. 11). In this assay, Balon homologues co-sedimented

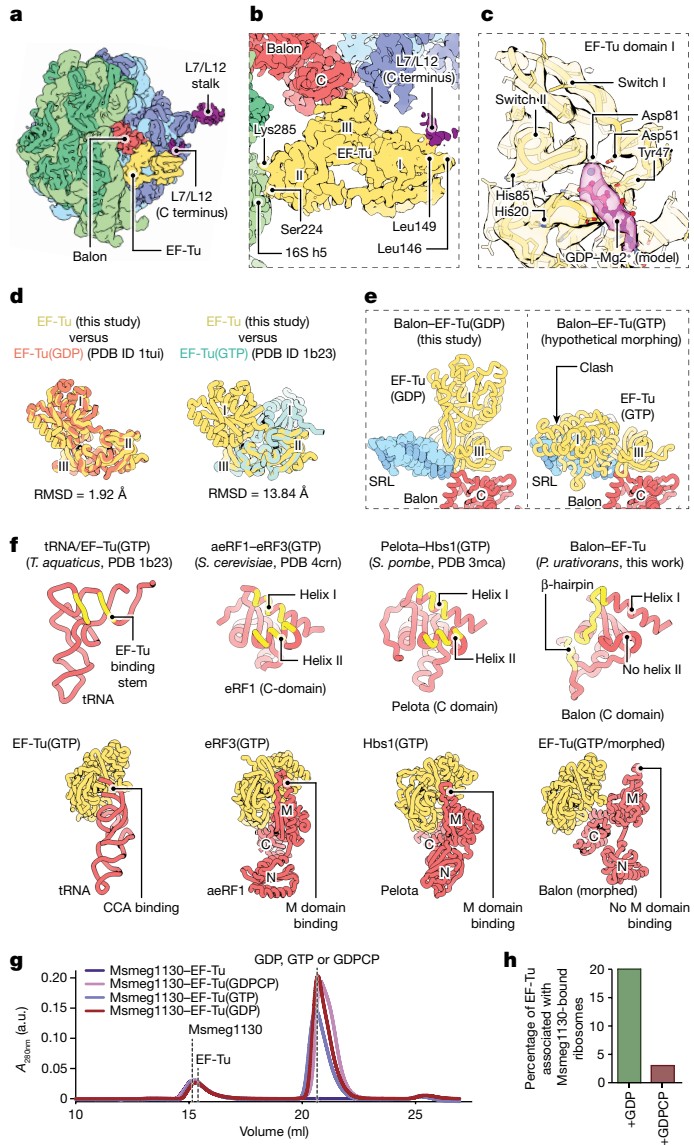

**Fig. 5 | Balon participates in ribosome hibernation in complex with the GDP-bound form of EF-Tu. a**, A cryo-EM map (filtered to 6 Å resolution) showing a *P. urativorans* ribosome bound to Balon and EF-Tu. **b**, A cryo-EM map focusing on the EF-Tu-binding sites illustrating that EF-Tu is attached to Balon-bound ribosomes through contacts with Balon, the C-terminal domain of the L7/L12 stalk (protein bL12) and the 16S rRNA helix h5. **c**, The structure of the EF-Tu nucleotide-binding domain and density consistent with the presence of GDP (the cryo-EM map is filtered to 4 Å resolution). **d**, A comparison of EF-Tu structures observed in this study and determined previously. EF-Tu molecules that are bound to Balon most closely resemble the GDP-bound conformation. **e**, Aligned structures showing that EF-Tu cannot adopt the GTP-bound conformation (observed in PDB code 1b23) while maintaining an interface with the C-terminal domain of Balon owing to a clash between EF-Tu domain I and the sarcin–ricin loop (SRL). **f**, Comparison of intramolecular interaction surfaces in four biological complexes, including EF-Tu–tRNA, eRF3–aeRF1, Hbs1–Pelota and EF-Tu–Balon. The upper panels highlight (in yellow) residues that recognize domain III of EF-Tu, or the EF-Tu homologues eRF3 and Hbs1. The lower panels compare complexes of EF-Tu(GTP)–tRNA, eRF3(GTP)–aeRF1 and Hbs1(GTP)–Pelota and the hypothetical complex of EF-Tu(GTP)–Balon, in which the Balon molecule is morphed to resemble the aeRF1 conformation in the eRF3(GTP)–aeRF1 complex. **g**, Size-exclusion chromatography shows that EF-Tu and Balon (Msmeg1130) do not form a stable complex even in the presence of GTP or GDPCP. **h**, Cryo-EM-based measurements of EF-Tu association with Msmeg1130-bound *M. smegmatis* ribosomes illustrate that EF-Tu effectively binds to Balon (Msmeg1130) in the ribosome only in the presence of GDP but not in the presence of GDPCP.

with ribosomes in the absence of EF-Tu, but EF-Tu was detected in the pellet only if Balon proteins were also present. The weak Balon–EF-Tu association is therefore stabilized in the context of the ribosome, possibly by the additional interactions that we observe between EF-Tu, the ribosomal L7/L12 stalk (protein bL12) and the 16S rRNA helix h5 (Fig. 5a,b).

To confirm the nucleotide dependency of the interaction, we reconstituted complexes of *M. smegmatis* ribosomes with the Balon homologue Msmeg1130 in the presence of EF-Tu(GDP) or EF-Tu(GDPCP) and analysed their structures by cryo-EM (Extended Data Figs. 5–8). In the presence of GDP, EF-Tu density was observed on 20% of particles containing Msmeg1130. By contrast, the addition of GDPCP strongly inhibited EF-Tu association with Msmeg1130-bound ribosomes, causing a seven-fold decrease in the proportion of EF-Tu-bound particles (2.9%; Fig. 5h). Unexpectedly, a low-resolution map from this subset of particles showed that EF-Tu still exhibited the open conformation, resembling a GDP-bound state (Extended Data Fig. 6). However, our EF-Tu(GDPCP) samples contained a small amount of co-purified GDP (Supplementary Fig. 9), consistent with previous reports on recombinant preparations of translational GTPases[45,46].

Collectively, our analysis showed that whereas aeRF1, Pelota and Balon bind to the A site in complex with EF-Tu, Balon uses a dissimilar EF-Tu recognition strategy and probably follows a different delivery mechanism to the ribosomal A site. Our data imply that this mechanism involves either Balon association with the ribosomal A site and a subsequent recruitment of EF-Tu(GDP) or Balon recruitment by EF-Tu(GDP) through the weak interactions between EF-Tu, Balon and the ribosomal L7/12 stalk. In either of these scenarios, Balon—unlike aminoacyl-tRNAs, aeRF1 and Pelota—does not engage with the GTP-bound form of EF-Tu, providing a possible explanation for why Balon does not interfere with protein synthesis during normal growth conditions, when cells contain abundant levels of GTP[47]. Therefore, in contrast to aminoacyl-tRNAs, aeRF1 and Pelota, Balon loading in the A site seems to bypass not only the step of mRNA codon verification but also the step of GTP hydrolysis, explaining how Balon is able to bind to nearly all cellular ribosomes during starvation and stress (Extended Data Fig. 9). This finding reveals a previously unknown biological activity of EF-Tu, illustrating that this protein participates not only in protein synthesis but also in ribosome hibernation.

## Discussion

### A novel family of hibernation factors

Here, by investigating an understudied psychrophilic bacterium under the conditions of stationary phase and cold shock, we have identified Balon: a new family of ribosome hibernation factors. Balon homologues are present in approximately 20% of studied bacteria and share structural similarity with archaeo-eukaryotic translation factors, rather than the two previously described bacterial ribosome hibernation factors. Balon possesses a unique ribosome-binding mechanism that allows it to associate with both vacant and actively translating ribosomes: a feature that sets it apart from all other known hibernation factors.

Our analysis of Balon structure shows that each globular domain of this protein has been functionally repurposed compared to the corresponding domains in aeRF1. Although both Balon and aeRF1 use their N-terminal domains to recognize the ribosomal decoding centre, Balon lacks the NIKS motif and bears an HP motif to bind the decoding centre without making any contact with the mRNA channel—thereby allowing Balon to bind ribosomes irrespective of the presence of an mRNA substrate. Both aeRF1 and Balon use their middle domains to engage with the peptidyl-transferase centre, but Balon lacks the GGQ motif and instead bears the bL27 trap that preserves the ribosomal active site in an inactive state. This probably prevents premature release of nascent peptides that could otherwise be toxic for the cell. Both

aeRF1 and Balon use their C-terminal domains to engage with EF-Tu, but Balon bears a unique β-hairpin insertion that seems to prevent Balon association with the GTP-bound EF-Tu, which probably limits Balon's interference with normal protein synthesis. These changes are likely to endow Balon with the ability to bind several functional states of the ribosome, as opposed to one specific state that is recognized by aeRF1. These structural changes provide an example of evolutionary specialization that allows aeRF1 to function as a termination factor in archaea and eukaryotes whereas Balon functions as a ribosome hibernation factor in bacteria.

Our data do not exclude the possibility that Balon has other activities in bacterial cells besides participating in ribosome hibernation. For example, bacterial cells respond to certain forms of stress by inducing the stringent response, which requires the binding of the stringent response factor RelA to the ribosomal A site[48,49]. Our finding that Balon occupies the ribosomal A site under two physiologically unfavourable conditions raises the possibility that Balon could also be an antagonist of the stringent response, independently of its role in ribosome hibernation. It will therefore be exciting to explore other possible roles for Balon in future, including the functional interplay between Balon and other A-site substrates.

### Elongating ribosomes can hibernate too

Many organisms, including *Psychrobacter* and *Mycobacteria* species, grow at substantially slower rates and produce proteins at slower rates (about 20 min to produce an average protein in *Mycobacteria)* compared to *E. coli* (about 15 s)[50]. This raises the question of how organisms with slower rates of translation can commence ribosome hibernation when exposed to a sudden change in their environment that does not give their ribosomes enough time to complete protein synthesis. Our discovery of Balon provides a possible answer and revises the model of ribosome hibernation, arguing against a single generalized mechanism inferred from studies in *E. coli*[1,2]. Until now, ribosomes were believed to enter hibernation only after completing the elongation cycle and transitioning into their vacant state to become accessible for hibernation factors. By contrast, we show that Balon has the unique property of binding to not only vacant ribosomes, but also ribosomes bound to mRNA and peptidyl-tRNA, illustrating that ribosomes can enter a hibernation state before terminating their protein synthesis. One possible benefit of this mechanism is that ribosomes can respond to stress faster, without having to wait for the mRNA translation cycle to complete. We reason that this more instantaneous mode of ribosome hibernation may be particularly important in slow-growing bacteria in which Balon would play an important role in pausing the substantial fraction of cellular ribosomes that cannot rapidly terminate their elongation when stress is encountered. This idea is consistent with our observation of an enrichment of ribosomes bound to both Balon and P-site tRNA in cells exposed to abrupt ice treatment, in comparison to those that underwent a gradual transition to stationary phase (Fig. 1b,c).

Notably, our finding of Balon in tRNA-containing ribosomes has important implications for cryo-EM studies of hibernating ribosomes. At present, cryo-EM data processing strategies typically exclude P-site tRNA-bound ribosomes during the early stages of particle classification when examining hibernating ribosomes. However, our study shows that this approach would result in an incomplete or even misleading understanding of ribosome hibernation. We propose that a more effective approach to identifying hibernating ribosomes should involve inspecting all ribosomal active sites for the presence of stress-response factors.

### EF-Tu in the bacterial stress response

At present, EF-Tu is thought to bind ribosomes only under normal growth conditions[49]. Our work, however, shows that EF-Tu may also engage with ribosomes during the bacterial stress response. The high conservation of the EF-Tu-binding β-hairpin among Balon homologues

implies that EF-Tu binding must play an important—if yet undefined—role. Our findings suggest that EF-Tu does not deliver Balon to the ribosomal A site using the same delivery mechanism as it uses for aminoacyl-tRNAs, aeRF1 and Pelota. In contrast to aminoacyl-tRNAs and aeRF1, Balon binds ribosomes in an mRNA-independent manner and does not detectably bind EF-Tu(GTP) (Fig. 5). Nonetheless, we cannot exclude that EF-Tu recruits Balon to the ribosome through weak interactions that cannot be detected through co-purification. Future work will be required to unravel the recruitment mechanism and any potential roles of EF-Tu in ribosome hibernation beyond recruitment of Balon. EF-Tu is the target of elfamycin antibiotics (for example, GE2270A and kirromycin); therefore, understanding the role of EF-Tu in hibernation could provide new insights into elfamycin activity[51]. Finally, the fact that EF-Tu associates with Balon and hibernating ribosomes only when it is bound to GDP suggests that the Balon–EF-Tu hibernation mechanism may represent a previously unknown mechanism for sensing intracellular levels of GTP, GDP or possibly ppGpp to allow bacteria to initiate ribosome hibernation in response to an energy deficit.

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

# Methods

## Production of *P. urativorans* biomass

As a model organism, we used the bacterium *P. urativorans*. Freeze-dried cells of *P. urativorans* were obtained from the American Type Culture Collection (ATCC 15174). The cell pellet was rehydrated in 15 ml of pre-chilled marine broth 2216 medium (Sigma-Aldrich) and incubated in a shaker (SciQuip Incu-Shake Mini) at 150 r.p.m. at 10 °C for 7 days, according to the American Type Culture Collection protocol. To isolate ribosomes corresponding to structures 1–3 (Extended Data Table 1), this culture was then used to inoculate 1 l of pre-chilled marine broth 2216 medium and incubated at 150 r.p.m. for 4 days at 10 °C until the culture reached an optical density at 600 nm ($OD_{600}$) of 0.272. The cells were then placed on ice for 10 min and centrifuged for 5 min at 4 °C and 5,000$g$, yielding approximately 1 g of cell pellet. To isolate ribosomes from stationary cells, *P. urativorans* cultures were allowed to reach the stationary phase ($OD_{600}$ of 1.5) and remain in this phase for 4 days before pelleting these cells for 5 min at 4 °C and 5,000$g$ and using this pellet for ribosome isolation.

## Ribosome isolation

To lyse the cells, the pellets were rapidly resuspended in 1 ml of buffer A (50 mM Tris-HCl pH 7.5, 20 mM magnesium acetate and 50 mM KCl), transferred to 2-ml microcentrifuge tubes containing approximately 0.1 ml of 0.5 mm zirconium beads (Sigma-Aldrich BeadBug), and disrupted by shaking for 30 s at 6.5 m s$^{-1}$ speed in a bead beater (Thermo FastPrep FP120 Cell Disrupter). The sample was then centrifuged for 5 min at 4 °C and 16,000$g$ to remove cell debris, and the resulting supernatant was collected and centrifuged for 1 min at 16,000 r.p.m. and 4 °C to remove the remaining debris. To analyse polysome profiles, we analysed 0.1 ml of crude *P. urativorans* lysates per time point, using 10–40% sucrose gradients in buffer A after 3 h of centrifugation at 35,000 r.p.m. and 4 °C in a SW41 rotor (Beckman Coulter). To isolate ribosomes for structural analysis, the cell lysate corresponding to 30 min of ice treatment was then mixed with PEG 20,000 (25% w/v) to a final concentration of 0.5% (w/v) and centrifuged for 5 min at 4 °C and 16,000$g$ to precipitate insoluble aggregates. Then, the supernatant was mixed with PEG 20000 (powder) to the final concentration of about 12.5% (w/v) and centrifuged for 5 min at 4 °C and 16,000$g$ to precipitate ribosomes. To monitor precipitation of ribosomes, we analysed lysates and their fractions using size-exclusion chromatography with Superdex Increase 200 10/300 in buffer A (Extended Data Fig. 1). The obtained ribosome-containing pellet was dissolved in 50 µl of buffer A, and the solution was passed twice through PD Spin Trap G-25 microspin columns (GE Healthcare) to clear crude ribosomes from small molecules. The obtained solution had an $OD_{260nm}$ of 34.89 and an $OD_{260nm/280nm}$ of 1.71, corresponding to a ribosome concentration of 512 nM. This solution was split into 10-µl aliquots and frozen at −20 °C for subsequent cryo-EM and mass spectrometry analyses.

## Mass spectrometry analysis of crude samples of *P. urativorans* ribosomes

For each measurement shown in supplementary datasets 1 and 2, a 10-µl aliquot of crude *P. urativorans* ribosome solution was reduced with 4.5 mM dithiothreitol and heated at 55 °C. The sample was alkylated with the addition of 10 mM iodoacetamide before proteolytic digestion with 0.2 µg Promega sequencing-grade trypsin and incubation at 37 °C for 16 h. The resulting peptides were desalted by Millipore C18 ZipTip, following the manufacturer's protocol, with final elution into aqueous 50% (v/v) acetonitrile. Desalted peptides were dried under vacuum before being resuspended in aqueous 0.1% trifluoroacetic acid (v/v) for LC–MS/MS.

Peptides were loaded onto a mClass nanoflow UPLC system (Waters) equipped with a nanoEaze M/Z Symmetry 100-Å C18, 5-µm trap column (180 µm × 20 mm, Waters) and a PepMap, 2 µm, 100 Å, C18 EasyNano nanocapillary column (75 µm × 500 mm, Thermo). The trap wash solvent was aqueous 0.05% (v/v) trifluoroacetic acid and the trapping flow rate was 15 µl min$^{-1}$. The trap was washed for 5 min before switching flow to the capillary column. Separation used gradient elution of two solvents: solvent A—aqueous 0.1% (v/v) formic acid; solvent B—acetonitrile containing 0.1% (v/v) formic acid. The flow rate for the capillary column was 330 nl min$^{-1}$ and the column temperature was 40 °C. The linear multi-step gradient profile was: 3–10% B over 7 min, 10–35% B over 30 min, 35–99% B over 5 min and then proceeded to wash with 99% solvent B for 4 min. The column was returned to initial conditions and re-equilibrated for 15 min before subsequent injections.

The nanoLC system was interfaced with an Orbitrap Fusion Tribrid mass spectrometer (Thermo) with an EasyNano ionization source (Thermo). Positive ESI-MS and MS2 spectra were acquired using Xcalibur software (v4.0, Thermo). Instrument source settings were: ion spray voltage—1,900 V; sweep gas—0 a.u.; ion transfer tube temperature—275 °C. MS1 spectra were acquired in the Orbitrap with 120,000 resolution, the scan range of $m/z$ 375–1,500, the AGC target of $4 \times 10^5$, and the maximum fill time of 100 ms. Data-dependent acquisition was carried out in top speed mode using a 1-s cycle, selecting the most intense precursors with charge states >1. Easy-IC was used for internal calibration. Dynamic exclusion was carried out for 50-s post precursor selection and a minimum threshold for fragmentation was set at $5 \times 10^3$. MS2 spectra were acquired in the linear ion trap with: scan rate—turbo; quadrupole isolation—1.6 $m/z$; activation type—HCD; activation energy—32%; AGC target—$5 \times 10^3$; first mass—110 $m/z$; maximum fill time—100 ms. Acquisitions were arranged by Xcalibur to inject ions for all available parallelizable time.

Peak lists in Thermo.raw format were converted to .mgf using MSConvert (v3.0, ProteoWizard) before submitting to database searching against the *P. urativorans* subset of the UniProt database (3 August 2022, 2,349 sequences; 769,448 residues)[52] appended with 118 common proteomic contaminants. Mascot Daemon (v2.6.0, Matrix Science) was used to submit the search to a locally running copy of the Mascot program (Matrix Science, v2.7.0). Search criteria specified: enzyme—trypsin; maximum missed cleavages—2; fixed modifications—carbamidomethylation of protein C termini; variable modifications—acetylation of protein N-termini, deamidation of Asn and Gln residues, N-terminal conversion of Gln and Glu to pyro-Glu, oxidation of Met and phosphorylation of Ser, Thr and Tyr residues; peptide tolerance—3 ppm; MS/MS tolerance—0.5 Da; instrument—ESI-TRAP. Peptide identifications were passed through the percolator algorithm to achieve a 1% false discovery rate assessed against a reverse database. The search data are summarized in supplementary datasets 1 and 2, for which molar percentages of each identified protein were calculated from Mascot emPAI values by expressing individual values as a percentage of the sum of all emPAI values in the sample, as previously described[53]. To calculate the relative abundance of each cellular protein before and after 30 min of ice treatment (as shown in supplementary dataset 1), their total spectrum counts in the ice-treated sample were divided by the corresponding total spectrum counts of the control (non-ice-treated) sample. An infinite value for a few proteins means that in the control sample we have not been able to detect evidence for a protein by spectral counting.

## Cryo-EM grid preparation and data collection for *P. urativorans* ribosomes

To prepare ribosome samples for cryo-EM analyses, 8–10-µl aliquots of crude ribosomes were thawed on ice and loaded onto glow-discharged (20 mA, 60 s or 90 s, PELCO easiGlow) Quantifoil grids (R1.2/1.3, 200 mesh, copper), using 2 µl of the sample per grid. The grids were then blotted for 1 or 2 s at 100% humidity (using blotting force −5) and vitrified using liquid nitrogen-cooled ethane in a Vitrobot Mark IV (Thermo Scientific). The grids were screened using Smart EPU (Thermo

Scientific) with a 200-kV Glacios electron cryo-microscope (Thermo Scientific) with a Falcon 4 detector located at the York Structural Biology Laboratory, University of York, UK. The dataset corresponding to structures 1–3 (Extended Data Table 1) was collected on a 300-kV Krios cryogenic electron microscope (Thermo Scientific) located at the electron Bio-Imaging Centre, Diamond Light Source, UK using the parameters detailed in Extended Data Table 1. A total of 9,637 micrograph videos were recorded in aberration-free image shift mode using defocus targets of −2.4, −2.1, −1.8, −1.5, −1.2 and −0.9 μm. The dataset corresponding to the stationary phase sample of *P. urativorans* ribosomes was collected using a 200-kV Glacios cryogenic electron microscope (Thermo Scientific) with a Falcon 4 detector located at the York Structural Biology Laboratory, University of York, UK. For each video, the grids were exposed to a total dose of 50 electrons Å$^{-2}$ across 5.65 s. A nominal magnification of ×150,000 was applied, resulting in a final calibrated object sampling of 0.94 Å pixel size. A total of 4,997 micrograph videos were recorded in aberration-free image shift mode using defocus targets of −1.4, −1.2, −1.0, −0.8 and 0.6 μm.

## Cryo-EM data processing for *P. urativorans* ribosomes

The cryo-EM dataset corresponding to structures 1–3 (Extended Data Table 1) was processed using RELION 3.1[54] as summarized in Extended Data Table 1 and Extended Data Fig. 1). In brief, to determine structures 1–3, a total of 180,467 particles were picked from 8,903 micrographs using the Laplacian of Gaussian picker (220–330 Å particle diameter; 0.6 s.d. threshold). Particle images were initially downscaled threefold and extracted in a 180-pixel box (2.169 Å effective pixel size). Four rounds of two-dimensional (2D) classification were carried out to clean the dataset, with 83,841 'good' particles selected for 3D refinement. These particles were rescaled to full size and extracted in a 540-pixel box. The initial 3D refinement generated a map at 3.5 Å resolution, using an initial 3D reference imported from a previous Glacios dataset that had been low-pass filtered to 60 Å. Heterogeneity was apparent at the Balon-binding site and decoding centre. At this point, contrast transfer function (CTF) refinement was carried out to account for beam tilt, trefoil fourth-order aberrations and magnification anisotropy. The CTF was estimated per particle, and the astigmatism was estimated per micrograph. Subsequent particle polishing and 3D refinement generated a map at 2.51 Å resolution, thereby providing the most accurate angular assignments for subsequent focused classification. Initial attempts at focused classification using masked classification without alignment were not successful, so signal subtraction was first carried out using masks to define Balon density and P-site density. This effectively separated particles into three groups corresponding to differential factor occupancy: ribosome (empty); structure 1 (*P. urativorans* ribosome–Balon–RaiA, with partial EF-Tu occupancy); and structure 2 (*P. urativorans* ribosome–Balon–tRNA–mRNA, with partial EF-Tu occupancy). The overall density for EF-Tu in the above structures 1 and 2 was weak, indicating substoichiometric amounts. Focused classification with signal subtraction was therefore carried out, first with a loose mask around the entire EF-Tu molecule. This revealed that about 44% of Balon-associated ribosomes were bound by EF-Tu. However, within this subset, the density for domain I of EF-Tu was poor, indicating conformational flexibility. Further signal subtraction was therefore carried out using a mask to isolate the density for EF-Tu domain I. Subtracted particles were recentred in a 200-pixel box and classification with local angular searches was carried out, revealing that domain I was present in two slightly rotated orientations with respect to domains II and III of EF-Tu. Particles corresponding to these two orientations of domain I were re-extracted and refined separately, leading to interpretable density at a local resolution of about 4 Å for EF-Tu domain I, at which point the details of the nucleotide-binding site became visible. Structure 3 represents the map with the best density for EF-Tu but is heterogeneous with

respect to tRNA or RaiA in the P site. In this structure, EF-Tu is also associated with the globular C-terminal domain of the L7/L12 stalk (protein bL12). A map filtered to about 6 Å shows this the most clearly (Fig. 5a and Supplementary Fig. 1). Finally, sharpened maps weighted by estimated local resolution were calculated. All reported estimates of resolution are based on the gold-standard Fourier shell correlation at 0.143, and the calculated Fourier shell correlation is derived from comparisons between reconstructions from two independently refined half-sets.

The cryo-EM dataset corresponding to ribosomes from stationary *P. urativorans* was processed using cryoSPARC v4.3.0[55] as summarized in Supplementary Fig. 2. In brief, a total of 909,391 particles were picked from 4,997 micrographs using the Blob picker (190–260 Å particle diameter). Particle images were extracted in a 400-pixel box (without downscaling). Five rounds of 2D classification were carried out to clean the dataset, with 80,882 'good' particles selected for ab initio reconstruction and subsequent homogeneous refinement. This generated a final map at 5.1 Å resolution, which was used for rigid-body docking of structure 1 to assess the presence of Balon in the ribosomal A site (Supplementary Fig. 2).

## Model building, refinement and deposition

The atomic models of *P. urativorans* ribosomes and the ribosome-binding proteins were produced using Coot v0.8.9.2[56] and AlphaFold[57]. As a starting model, we used the atomic model of ribosomal proteins generated by AlphaFold2 and the atomic model of rRNA from the coordinates of *T. thermophilus* ribosomes (PDB code 4y4o). These rRNA and protein models were morph-fitted into the cryo-EM maps using ChimeraX 1.4[58] and Phenix 1.20.1[59] and then rebuilt using Coot on the basis of the information about the genomic sequence of *P. urativorans* (RefSeq GCF_001298525.1). In the ribosome complex with Balon, mRNA and tRNA, the mRNA molecule was modelled as poly-U, and the tRNA molecule was modelled as $U_{1-72}A_{73}C_{74}C_{75}A_{76}$.

The density corresponding to Balon was initially identified as a non-ribosomal protein, which was initially modelled as a poly-alanine chain to determine its backbone structure. This poly-alanine backbone model was then used as an input file for a search of proteins with similar fold in the PDB using the National Center for Biotechnology Information tool for tracking structural similarities of macromolecules, Vast[60]. This search identified the archaeal protein aeRF1 from *Aeropyrum pernix* as the most similar known structure to Balon, suggesting that Balon is a bacterial homologue of aeRF1 (Supplementary Table 1). We therefore searched for *P. urativorans* proteins that have a similar sequence to that of *A. pernix* aeRF1. Using three iterations of Markov model-based search with HHMER[61] in the UniProt database, we found that *P. urativorans* encodes a hypothetical protein (UniProt ID A0A0M3V8U3) with sequence similarity to aeRF1 and Pelota. This protein, which we termed Balon, had a sequence that matched the cryo-EM map and was used to create its atomic model. The resulting atomic structures of Balon in complex with the ribosome, RaiA and EF-Tu or Balon in complex with the ribosome, tRNA and mRNA were then refined using Phenix real-space refinement, and the refined coordinates were validated using MolProbity[62].

## Purification and cryo-EM analysis of *M. smegmatis* ribosomes and their complexes with Msmeg1130, Rv2629 and EF-Tu

*M. smegmatis* 70S ribosomes, isolated from strain mc²155, were prepared as previously described[63]. The full-length Rv2629 sequence was PCR amplified from *M. tuberculosis* H37Rv genomic DNA and the full-length *M. smegmatis* EF-Tu and Msmeg1130 were amplified from *M. smegmatis* mc²155 gDNA and cloned into the pET28a plasmid with a 6×His-SUMO tag. *E. coli* BL21 (DE3) Star cells were transformed with the constructs and grown at 37 °C in the LB medium with kanamycin. The expression of the proteins of interest was induced with

isopropyl-β-D-thiogalactoside once the culture reached an $OD_{600nm}$ of about 0.6. The purification protocol for mycobacterial proteins included a two-step Ni-affinity (HisTrap HP column), ion exchange (HiTrap Q HP or Source 15Q) and size-exclusion chromatography (Superdex 200 16/600). Rv2629, Msmeg1130 and *M. smegmatis* EF-Tu were flash frozen in liquid nitrogen and stored in 20 mM HEPES-KOH pH 7.5, 10 mM $MgCl_2$, 200 mM KCl, 300 mM L-arginine and 1 mM dithiothreitol.

*M. smegmatis* ribosome complexes (20 μl) with Msmeg1130 or Rv2629 were prepared by incubating 2 μM *M. smegmatis* mc$^2$155 70 S ribosomes with 30 μM Msmeg1130 or Rv2629 in 1× buffer B (20 mM HEPES-KOH pH 7.5, 60 mM KCl, 10 mM $MgCl_2$, 1 mM DTT) for 15 min at room temperature. Ribosome complexes with Msmeg1130 and EF-Tu were prepared by pre-incubating 60 μM EF-Tu with 1 mM GDP or 1 mM GDPCP for 5 min at 37 °C and mixed with *M. smegmatis* mc$^2$155 70S ribosomes to final concentrations of 2 μM ribosomes, 30 μM Msmeg1130 and EF-Tu followed by a 15-min incubation at room temperature.

### Cryo-EM grid preparation and data collection for *M. smegmatis* ribosomes

Plasma-cleaned Quantifoil (R2/1, 200 mesh, gold) grids (Electron Microscopy Sciences) were used for sample application. Grids were blotted in 85% humidity at room temperature for 22 s and plunge frozen in liquid nitrogen-cooled ethane using the Leica EM GP2 cryo-plunger. Then, cryo-EM micrographs were recorded with a Titan Krios G3i electron microscope (300 kV) equipped with FalconIII (ThermoFisher) and K3 (Gatan) direct electron detectors. A total of 11,031 (dataset 4, Extended Data Table 1 and Supplementary Fig. 3), 9,546 (dataset 5, Supplementary Fig. 4) and 10,161 (dataset 6, Extended Data Fig. 5) micrograph videos were acquired in the counting mode with a pixel size of 0.839 Å per pixel (K3) or fast integrating mode with 0.85 Å per pixel (FalconIII). On the basis of the relative ice thickness, patch CTF fit, length and curvature of motion trajectories, 9,440, 9,438 and 9,465 micrographs were selected for further processing. For all datasets with *M. smegmatis* 70S ribosomes, the particles were picked using the circular 'blob' picker in cryoSPARC and filtered on the basis of defocus-adjusted power and pick scores. Particles were then subjected to one (dataset 5) or two (datasets 4 and 5) rounds of reference-free 2D classification. The selected particles were used to generate ab initio volumes that were sorted using 'heterogeneous refinement'. Selection of the classes with Msmeg1130 or Rv2629 or Msmeg1130–EF-Tu bound to *M. smegmatis* 70S was carried out using 3D classification analysis, further classified and polished by focused 3D variability analysis (Msmeg1130EF-Tu datasets) with a spherical mask around EF-Tu. This approach allowed us to remove 'bad' or noisy particles, re-extract to full-size (512-pixel box) particles with solid density for factors and carry out non-uniform with CTF refinements, which yielded the final reconstructions for *M. smegmatis* 70S complexes. The model was assembled from individual parts. The non-rotated *M. smegmatis* 70S ribosome model (PDB code 5o61) structure with P-site tRNA in the active site[63] was rigid-body fitted into the 3.0-Å charge density maps using UCSF Chimera 1.14[64]. The models predicted by AlphaFold2[65,66] for Rv2629 and Msmeg1130 were docked into the density maps and adjusted in Coot v0.8.9.2[56]. mRNA was modelled as poly-U and tRNA$^{Phe}$ was modelled in the P site. The complete model was refined using five cycles of real-space refinement in Phenix 1.19.2[59].

A total of 11,846 micrographs (GDPCP dataset, Extended Data Fig. 6) were selected for further processing. The model predicted by AlphaFold2[65,66] for EF-Tu was used for the fitting into the density maps. The *Thermus aquaticus* EF-Tu(GDP) model (PDB code 1tui)[42] was used for fitting domain I and switches I and II into the EF-Tu density map. Local refinement combined with particle subtraction produced higher-quality maps for EF-Tu in complexes formed in the presence of GDP (Extended Data Fig. 5) and GDPCP (Extended Data Fig. 6).

### Evolutionary analysis of Balon

To assess phylogenetic distribution of Balon in bacterial species, we carried out three iterations of homology search using the sequence of Balon from *P. urativorans* (UniProt ID A0A0M3V8U3) as an input for a profile hidden Markov model-based analysis with HMMER. For each search iteration, we used the following search options: -E 1 --domE 1 --incE 0.01 --incdomE 0.03 --seqdb uniprotrefprot. The resulting dataset was reduced first by removing protein sequences that lacked information about their Phylum (21 sequences), then by removing sequences that were shorter than 300 amino acids as they typically lacked one or two of the three domains of Balon/aeRF1 (which included 806 sequences), then by removing sequences that were annotated as a protein fragment (34 sequences), and finally by removing duplicated sequences (31 sequences). This resulted in the dataset including 1,896 sequences of Balon homologues from 1,565 bacterial species (supplementary dataset 3).

To gain insight into a possible evolutionary origin of Balon from the archaeo-eukaryotic family of aeRF1 proteins, we carried out a complementary analysis in which we searched for bacterial homologues of the archaeal aeRF1 using three iterations of HMMER. As an input for the first iteration, we used the sequence of aeRF1 from the archaeon *A. pernix* (UniProt ID Q9YAF1), which we identified as being one of the closest structural homologues of Balon. For each iteration, we used the database of reference proteomes restricted to the bacterial domain of life, using these search options: -E 1 --domE 1 --incE 0.01 --incdomE 0.03 --seqdb uniprotrefprot. The resulting dataset was reduced first by removing sequences lacking information about their phylum (21 sequences), then by removing sequences that were lacking at least one of the three domains of aeRF1 proteins (sequences shorter than 300 amino acids, which included 1,422 sequences), then by removing sequences annotated as a protein fragment (5 sequences), and finally by removing duplicated sequences (104 sequences). This resulted in the dataset of 1,617 sequences of bacterial aeRF1 homologues from 1,353 bacterial species (supplementary dataset 4).

To map the identified Balon homologues on the tree of life, we combined the results of the previous searches in supplementary datasets 3 and 4 and removed repetitive entries, which resulted in a dataset of 1,898 protein sequences from 1,572 bacterial species (supplementary dataset 5). We then aligned the combined sequences using Clustal Omega[67] with default parameters, which resulted in a multiple sequence alignment (supplementary dataset 6) and a phylogenetic tree (supplementary dataset 7). To compare phylogenetic distribution of Balon, RMF and RaiA-type hibernation factors, we repeated the homology search using HMMER (with the same parameters as for our Balon homologues searches) for RaiA (using the *E. coli* sequences of RaiA as an input; supplementary dataset 8) and RMF (using the *E. coli* sequence of RMF as an input; supplementary dataset 9).

### Generation of Balon knockout strains

The gene knockout in *P. arcticus* 273-4 (DSMZ 17307, ref. seq. NC_007204.1) was generated on the basis of a suicide-vector-based approach originally developed previously[68]. In brief, the suicide vector pBC19 was constructed on the basis of the design of pJK100. Regions of 500–600 base pairs serving as homologies for the targeted deletion were PCR amplified from genomic DNA isolated from the respective strains. A third fragment containing a kanamycin resistance cassette flanked by two *loxP* sites was amplified from the pCM184 (Addgene no. 46012) vector. The three fragments were each amplified using primers carrying overlaps necessary for Gibson Assembly (NEBuilder HiFi DNA Assembly Master Mix, NEB). These three fragments were then assembled together and fused with the pKNOCK-Tc vector (Addgene no. 46259) and digested with EcoRI and KpnI using Gibson Assembly. The pKNOCK-Tc backbone carries a R6Kγ origin of replication requiring the *pir* gene for

replication; therefore, the Gibson reactions were first transformed into the *E. coli* MDS42pir strain[69]. A desired clone was then identified and purified and transformed into the conjugative strain *E. coli* BW29427 that is auxotrophic for diaminopimelic acid (DAP). This donor strain was then grown at 37 °C in LB broth medium supplemented with 50 mg ml⁻¹ kanamycin, 20 mg ml⁻¹ tetracycline and 100 mg ml⁻¹ DAP overnight and mixed with cultures of *P. arcticus* recipients, both grown for 48 h at 20 °C in LB broth. Aliquots of 1 ml of the cultures were centrifuged (2 min, 4,000 r.p.m.) and washed twice in PBS (8 g l⁻¹ NaCl, 0.2 g l⁻¹ KCl, 1.44 g l⁻¹ Na₂HPO₄, 0.24 g l⁻¹ KH₂PO₄, pH 7.4) and finally resuspended in 1 ml PBS. The cultures were then mixed gently at ratios of 100 ml donor to 100 ml recipient and 100 ml donor to 400 ml recipient and spot plated onto LB agar plates containing 100 mg ml⁻¹ DAP and grown for 24 h at 20 °C. The grown conjugation mixtures were then scraped and suspended in 1 ml LB broth, after which 100 ml of the suspension and 10× and 100× dilutions were spot plated onto LB agar plates containing 50 mg ml⁻¹ kanamycin and grown at 20 °C. Putative conjugant colonies became visible after 72–96 h. These were picked and checked for their sensitivity against 20 mg ml⁻¹ tetracycline. Tetracycline-sensitive colonies (with presumably the kanamycin resistance cassette inserted into the targeted genomic site) were re-streaked onto LB agar plates supplemented with 50 mg ml⁻¹ kanamycin. Perturbation of the target gene was then validated using PCR, with one primer annealing outside of the genomic homology region, and the other to the kanamycin resistance gene. In this manner, the expected fragment was produced only from colonies where the resistance cassette was genomically inserted into the desired locus. These PCR fragments were then sequence-verified using Sanger sequencing. Additionally, plasmid pBC16 was also constructed (on the basis of pJK100) carrying just the *loxP*-kanamycin resistance marker-*loxP* cassette cloned into the pKNOCK-Tc backbone carrying multi-cloning sites flanking the knock-in cassette. This plasmid can be used to clone homology cassettes using traditional restriction endonuclease-based cloning. To assess the growth of the produced strains, they were incubated in the BioTek TS800 microplate reader while measuring their growth using Agilent BioTek Gen 5 software.

## Analysis of *P. arcticus* rRNA

Cell pellets of log phase and stationary phase cells of wild-type *P. arcticus* and *P. arcticus* with the Balon-coding gene deleted were obtained by centrifuging 120 ml of an actively growing *P. arcticus* culture and 25 ml of an 11-week-old culture of *P. arcticus* at 5,000*g* for 5 min. The biomass of each of the four resulting cell samples was taken in the same quantity of 170 mg and resuspended in 750 μl of TRIzol reagent (Invitrogen) with 20 μl of zirconia beads and vortexed for 5 min to lyse the cells. Then, 150 μl of chloroform was added to the lysed cells and vortexed for 2 min. After 15 min, the lysates were centrifuged for 15 min at 13,000 r.p.m., resulting in phase separation of the samples. The upper aqueous phase, containing the RNA, was transferred to a clean Eppendorf tube, and mixed with 350 μl of isopropanol. After inverting the samples and allowing them to stand for 10 min, the samples were centrifuged for 10 min at 13,000 r.p.m. The resulting supernatant was decanted, and the RNA pellets were washed with 1 ml of 70% ethanol and centrifuged for 15 min at 13,000 r.p.m. After decanting the ethanol, the pellets were dried in a speed vacuum and resuspended in 18 μl of RNase-free water. The resuspended RNA was incubated at 55 °C for 5 min to allow for complete resuspension. Then, 2 μl of each sample or 1 μl RiboRuler Low Range RNA ladder (Thermo Scientific) was mixed with 5 volumes of 5× glyoxal loading dye (61.2% dimethylsulfoxide (v/v), 20.4% glyoxal (v/v), 12.2% 10× BPTE buffer (300 mM Bis-Tris, 100 mM PIPES (piperazine-*N*,*N*′-bis(2-ethanesulfonic acid)), 10 mM EDTA) (v/v), 4.8% glycerol (v/v)) with 0.2 mg ml⁻¹ ethidium bromide and incubated at 55 °C for 1 h. Glyoxalated RNA samples were separated on a 1.2% agarose 1× BPTE gel before being visualized on a PhosphorImager (Typhoon FLA9000; GE Healthcare).

## Reporting summary

Further information on research design is available in the Nature Portfolio Reporting Summary linked to this article.

## Data availability

Supplementary datasets 1–9 are available at https://figshare.com/s/374a95769c5f7e9cdc04. The atomic coordinate files for structures 1–6 have been deposited in the Protein Data Bank (PDB) with the accession codes 8RD8, 8RDV, 8RDW, 8V9J, 8V9K and 8V9L, and the associated cryo-EM maps were deposited to the Electron Microscopy Data Bank (EMDB) with the accession codes EMD-19067, EMD-19076, EMD-19077, EMD-43074, EMD-43075, EMD-43076, EMD-43077 and EMD-43078.

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

**Acknowledgements** We thank B. Javid, Z. Lightowlers, B. van der Berg and H. Murray for comments on the manuscript; J. Turkenburg and S. Hart for work supporting the York cryo-EM facility; and A. Kereszt for providing the conjugative strain *E. coli* BW29427. This work was financed by the Newcastle University NUORS 2021 Award (to K.H.-B.), the James W. McLaughlin Fellowship Fund (to M.Yu.R.), the Medical Research Council (MR/N013840/1 to C.L.E.), the Biotechnology and Biological Sciences Research Council UK (BB/T008695/1 to C.R.B.), a UKRI Future Leader Fellowship (MR/T040742/1 to J.N.B.), the Lendület (Momentum) Program of the Hungarian Academy of Sciences (LP2022-4/2022a to B.C.), a National Institutes of Health grant (R01GM136936 to M.G.G.), a Welch Foundation grant (H-2032-20230405 to M.G.G.), a Wellcome Trust & Royal Society Sir Henry Dale Fellowship (221818/Z/20/Z to C.H.H.) and the Royal Society (RGS/R2/202003 to S.V.M.). This project was undertaken on the NSBL Cluster and the Viking Cluster, which are high-performance compute facilities provided by Newcastle University and the University of York, respectively. We are grateful for computational support from the University of York High Performance Computing service, Viking and the Research Computing team, and support from the Newcastle University Structural Biology Laboratory. We also acknowledge the York cryo-EM facility supported by the Wellcome Trust (206161/Z/17/Z) and the York Centre of Excellence in Mass Spectrometry that was created with a capital investment through Science City York and supported by the Engineering and Physical Sciences Research Council (EP/K039660/1; EP/M028127/1) and Yorkshire Forward with funds from the Northern Way Initiative. We also acknowledge Diamond UK for access to and support of the cryo-EM facilities at the UK national electron Bio-Imaging Centre, proposal BI28576, financed by the Wellcome Trust, the Medical Research Council and the Biotechnology and Biological Sciences Research Council. We are grateful to M. Sherman for advice and support; K.-Y. Wong and J. Perkyns for computational support; and to the Sealy and Smith Foundation for supporting the Sealy Center for Structural Biology at the University

of Texas Medical Branch. For the purpose of open access, the authors have applied a CC BY public copyright license to any author accepted manuscript version arising from this submission.

**Author contributions** K.H.-B. and S.V.M. conceived the project. K.H.-B. isolated and characterized *P. urativorans* ribosomes using biochemistry and cryo-EM. M.Yu.R. isolated and characterized mycobacterial ribosomes and proteins using biochemistry and cryo-EM. C.H.H., J.N.B. and A.B. assisted with cryo-EM analysis and data processing. K.H.-B., C.L.E. and S.V.M. produced the atomic model of *P. urativorans* ribosomes. M.Yu.R. and M.G.G. built the atomic model of mycobacterial ribosomes. R.S., J.P.R.C. and P.M. contributed to the Balon genetic knockout design. B.C. created knockout in *Psychrobacter* species. C.D. produced ribosome samples from stationary *P. urativorans*. C.R.B. assisted with the search for Balon homologues. K.H.-B. and C.S. carried out rRNA isolation and biochemical analysis. C.H.H. processed *P. urativorans* cryo-EM data and identified functional states of ribosomes using particle classification. K.H.-B., M.Yu.R., M.G.G., C.H.H. and S.V.M. analysed the data and wrote the manuscript.

**Competing interests** The authors declare no competing interests.

**Additional information**
**Correspondence and requests for materials** should be addressed to Matthieu G. Gagnon, Chris H. Hill or Sergey V. Melnikov.

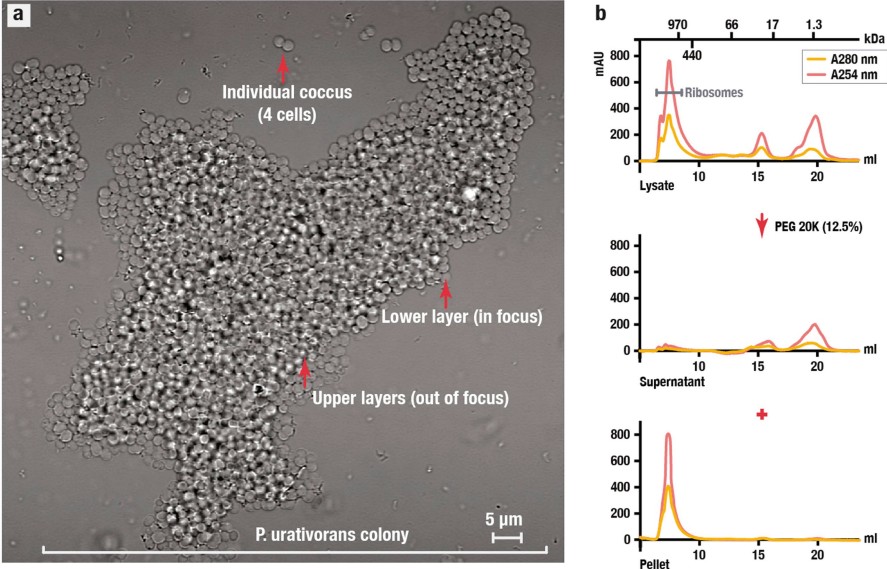

**Extended Data Fig. 1 | Ribosome isolation from the cold-adapted bacterium *P. urativorans*.** (**a**) A brightfield microscopy image of *P. urativorans* cells (the imaging was repeated three times independently showing similar results) used in this study shows a colony comprising approximately 1,000 cells. This colony was isolated from an actively growing liquid culture of *P. urativorans* ($OD_{600}$ ~ 0.2) by transferring 10 μL of a cell suspension onto an agar bed. Unlike many common model bacteria, such as *E. coli*, *P. urativorans* is not a unicellular organism but rather a multicellular organism[1]. Each individual cell of this bacterium is organized in a so-called coccus, which contains two, four, or more cells that are surrounded by a thick cell wall and internally divided by strongly developed cross-walls. Most of these cocci further self-assemble into larger cellular aggregates, like the one shown here. These aggregates typically comprise a few dozen to a few hundred cells arranged into carpet-like monolayers, with several monolayers attached to each other. This morphology, along with the presence of bright pigments in *P. urativorans* cells, makes this species unsuitable for fluorescence microscopy or cytometry studies. (**b**) Size-exclusion chromatography profiles illustrate the lysate fractionation strategy used in this study. Before PEG 20,000 fractionation, the lysate contains particles of various sizes (the upper panel). However, once the PEG 20,000 is added to the lysate, this causes selective precipitation of large particles, including ribosomes, as evident from their disappearance from the soluble fraction (the middle fraction) and their accumulation in the pellet (the lower fraction). Thus, by precipitating the content of cell lysates with PEG 20,000 (12.5%), we were able to achieve nearly complete isolation of *P. urativorans* ribosomes.

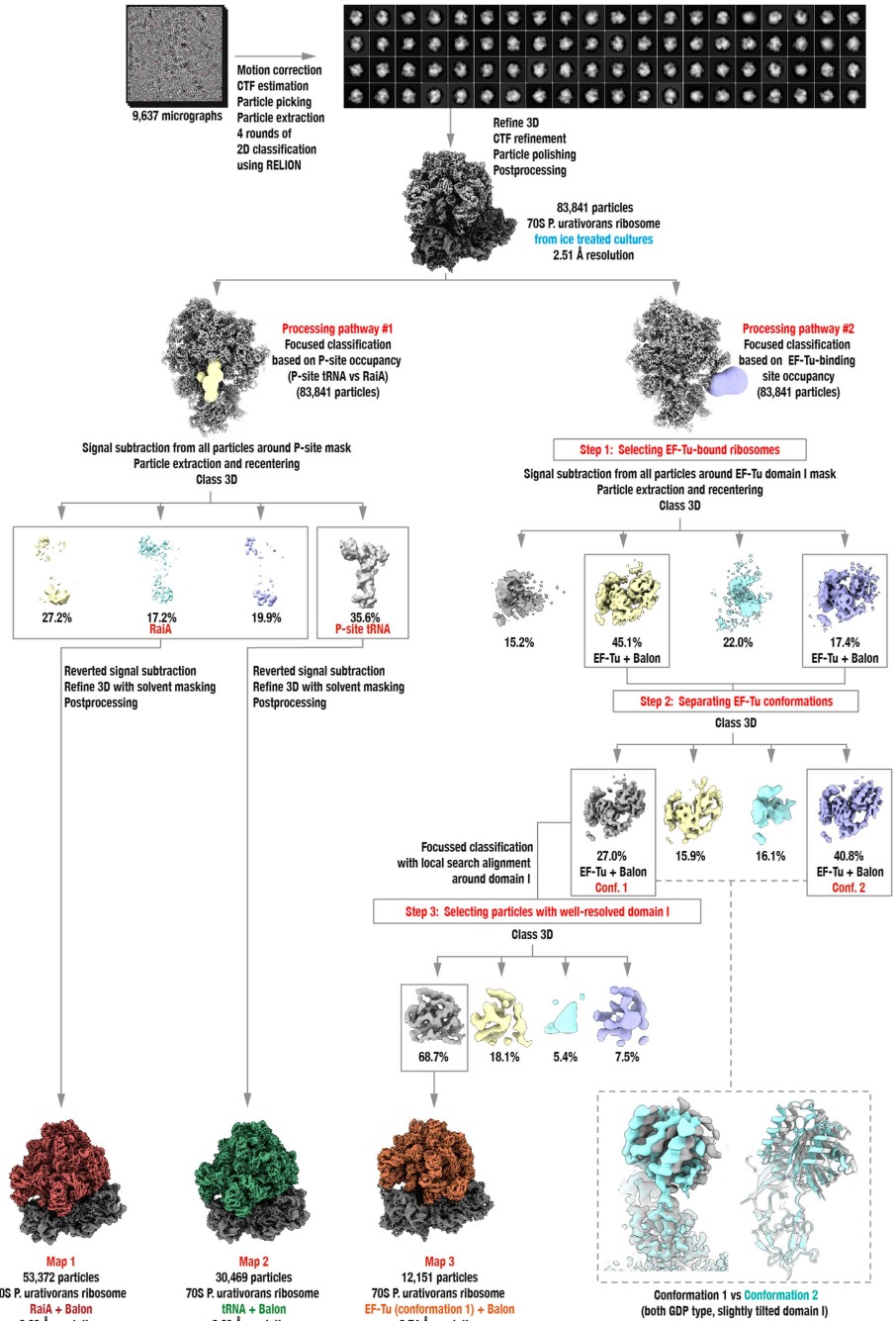

**Extended Data Fig. 2 | Cryo-EM data processing workflow for the reconstruction of *P. urativorans* ribosomes and focused classification analysis of Balon-bound ribosomes (Dataset 1 – ice shock), corresponding to *P. urativorans* ribosomes Structures 1–3.** The pipeline shows a representative micrograph at 165,000x, 2D classes, 3D reconstructions and major steps of data processing using RELION 3.1.

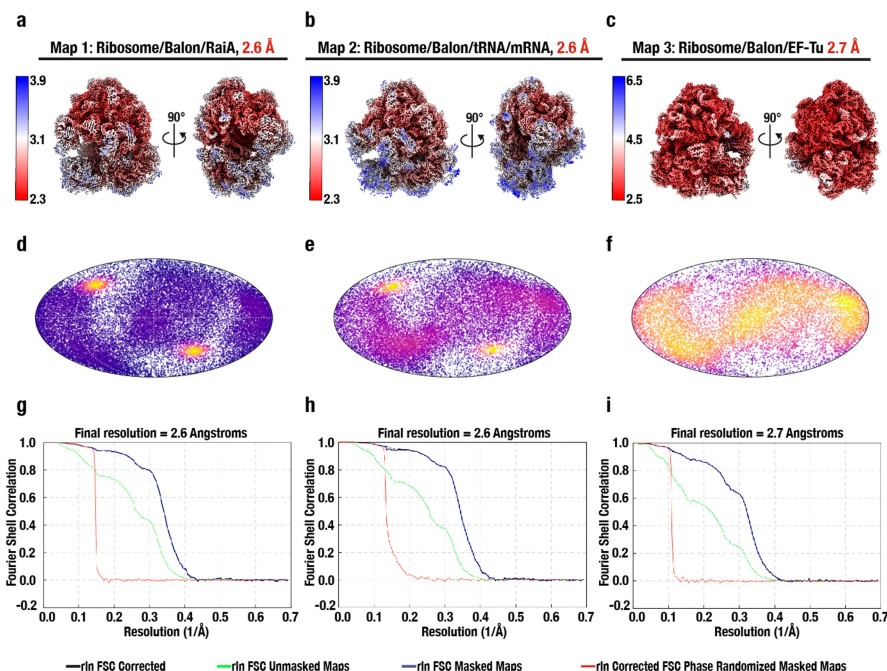

**Extended Data Fig. 3 | Validation of the cryo-EM maps for *P. urativorans* ribosome (ice shock).** (**a**–**f**) Panel descriptions refer to the Ribosome/Balon/RaiA map (left), Ribosome/Balon/tRNA/mRNA map (centre) and Ribosome/Balon/EF-Tu map (right) as indicated at the top of the figure. (**a**–**c**) Final cryo-EM maps, surface coloured by estimated local resolution as indicated in the heatmap key. Two orthogonal views are shown to illustrate two opposing sides of the ribosome particle. (**d**–**f**) Angular distribution plot of particles in the final reconstructions, shown as a Mollweide projection. (**g**–**i**) Gold-standard Fourier shell correlation (FSC) curves for final maps generated by RELION post-processing. Masked (blue), unmasked (green), and phase-randomised masked (red) plots are shown.

**Bacterial ribosomes from P. urativorans bound to mRNA, peptidyl-tRNA and Balon**

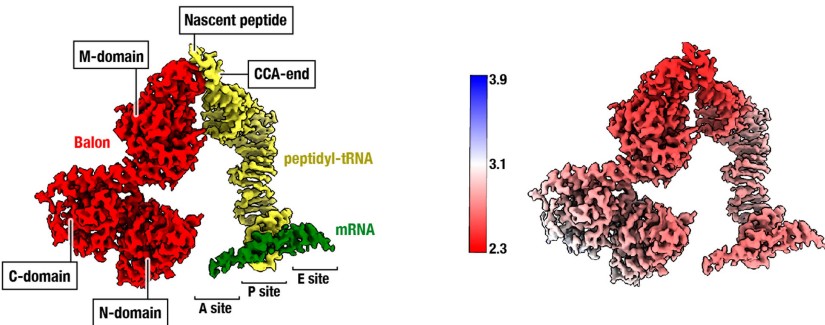

**Bacterial ribosomes from P. urativorans bound to the hibernation factor RaiA and Balon**

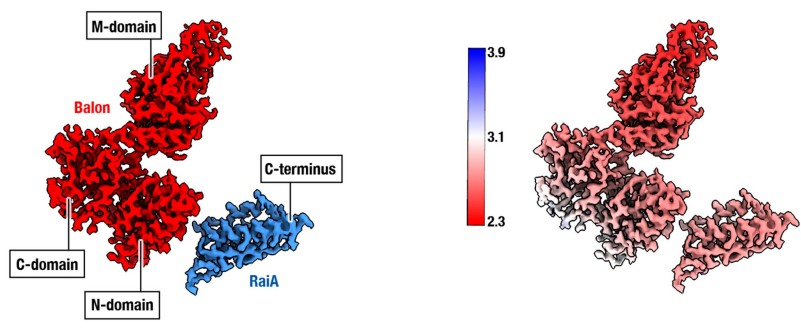

**Bacterial ribosomes from P. urativorans bound to EF-Tu and Balon**

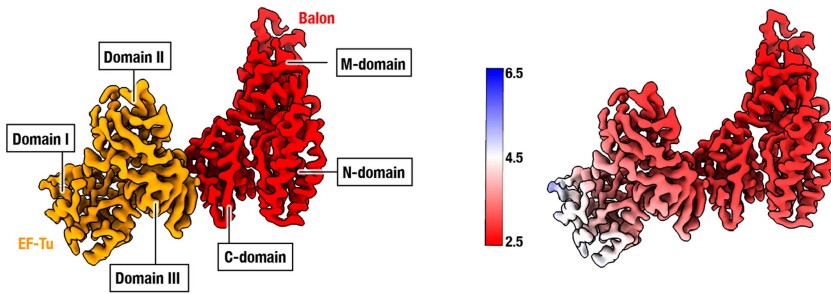

**Extended Data Fig. 4 | Cryo-EM maps for molecules that bind ribosomes concurrently with Balon (ice shock).** The cryo-EM maps show ribosomal ligands in two distinct classes of ribosomes in our dataset, with panels on the left coloured by chain and panels on the right coloured by local resolution. The first class represents the ribosome complex that contains heterologous mRNA, heterologous peptidyl-tRNA, and Balon. The second class represents the ribosome complex that comprises Balon and the hibernation factor RaiA. The third class corresponds to ribosome particles that bind Balon in complex with EF-Tu. The three domains of Balon are labelled as N-domain (N-terminal domain), M-domain (middle domain) and C-domain (C-terminal domain).

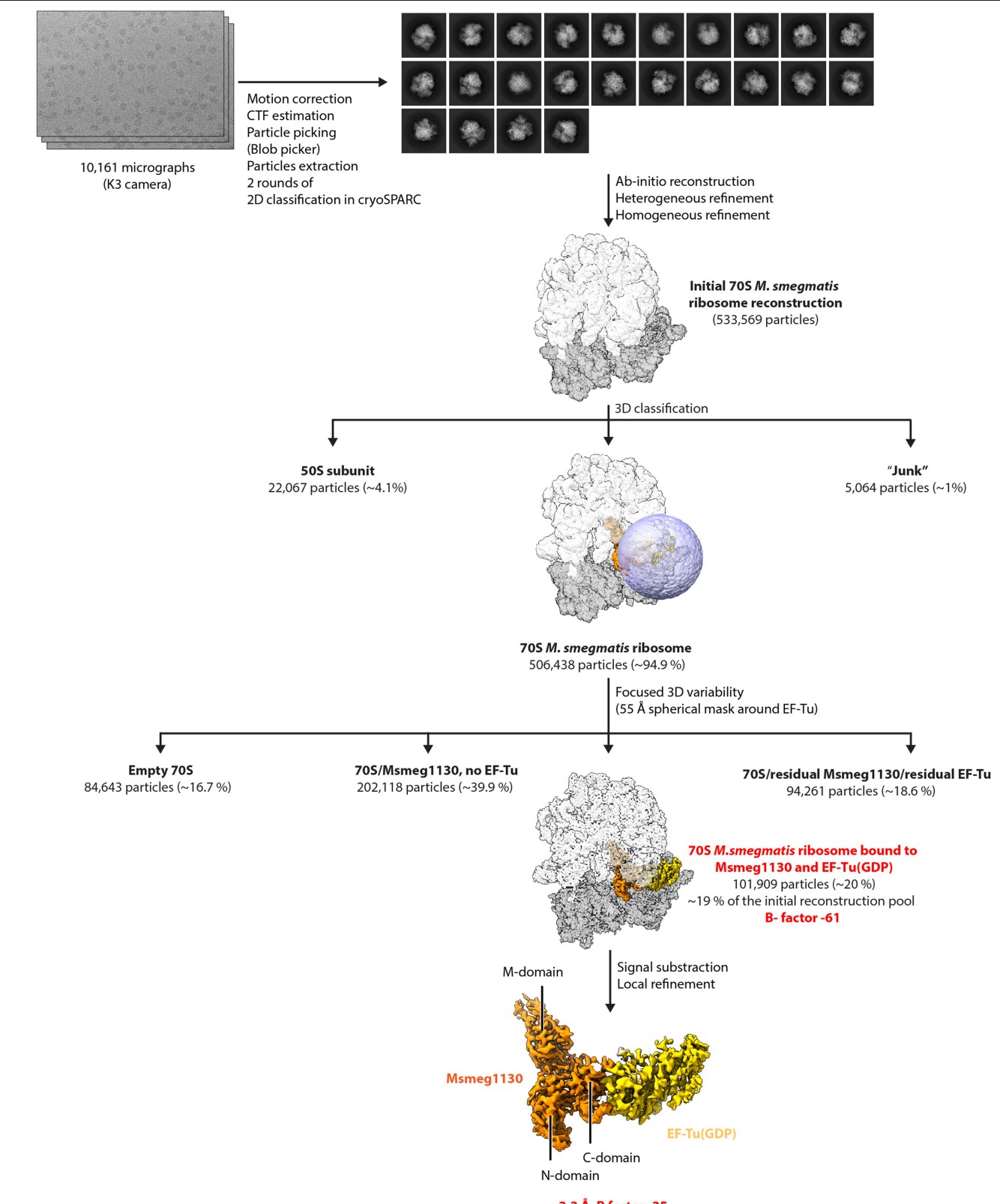

**Extended Data Fig. 5 | Cryo-EM data processing workflow using CryoSPARC for *M. smegmatis* ribosomes, corresponding to Structure 6 that comprises 70S ribosome in complex with Msmeg1130 and EF-Tu in the presence of GDP.**

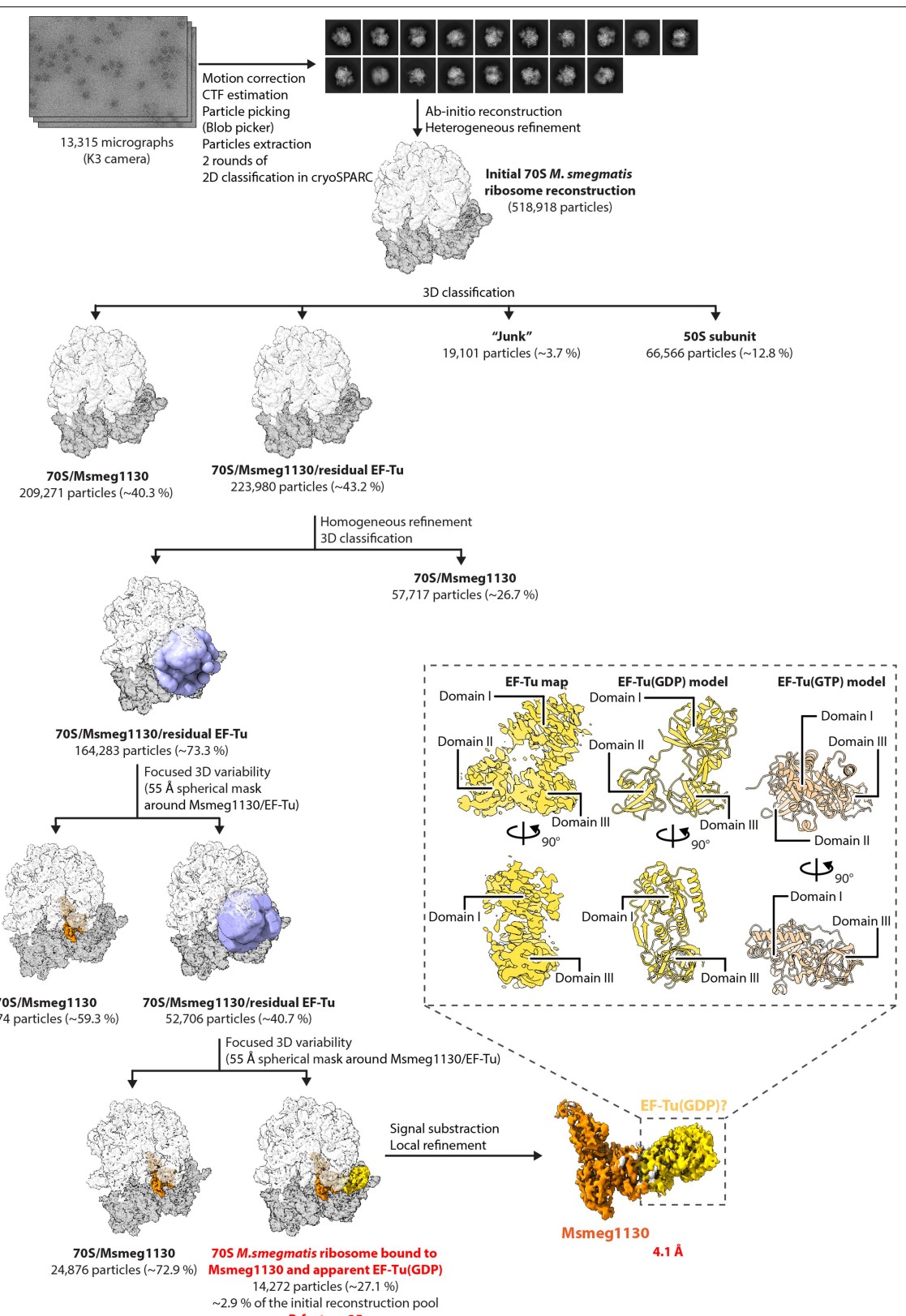

**Extended Data Fig. 6 | Cryo-EM data processing workflow using CryoSPARC for *M. smegmatis* ribosomes in complex with Msmeg1130 and EF-Tu in the presence of GDPCP.** The inset compares the EF-Tu signal with EF-Tu structures in the GTP and GDP states and shows that the EF-Tu density is not sufficient to build its molecular model but is sufficient to observe the overall conformation of the EF-Tu molecule. This conformation is consistent with the GDP-bound conformation of EF-Tu rather than the GTP-bound conformation, which likely stems from the residual amounts of GDP copurifying with our recombinant EF-Tu sample.

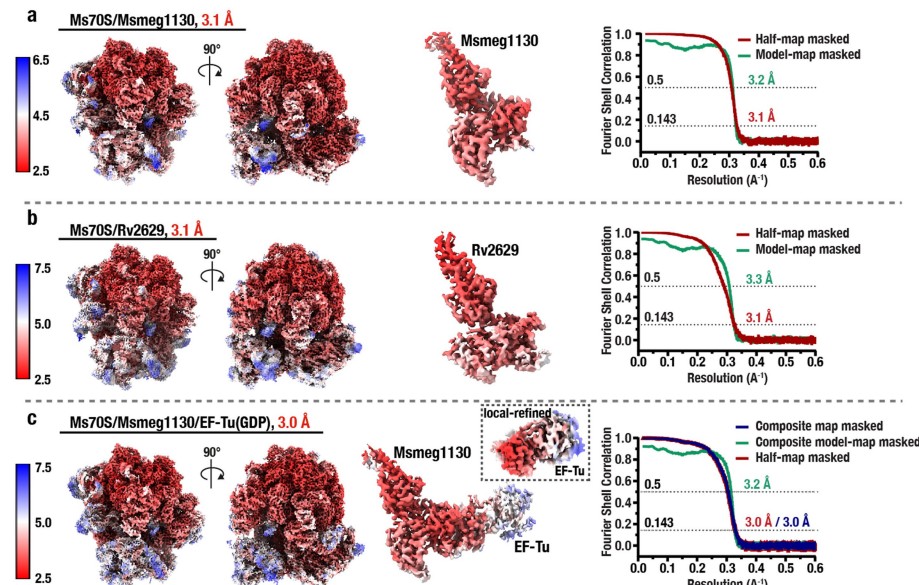

**Extended Data Fig. 7 | Validation of the cryo-EM maps for *M. smegmatis* ribosomes.** Local resolution estimation and gold-standard Fourier Shell Correlation (FSC) validation. Final cryo-EM maps for **Structure 4** (**a**) 70 S/Msmeg1130, **Structure 5** (**b**) 70 S/Rv2629, and **Structure 6** (**c**) 70 S/Msmeg1130/EF-Tu(GDP), surface coloured by estimated local resolution as indicated in the heat map scale. Two orthogonal views are shown. The gold-standard FSC curves of each half-map (red), using a 'soft mask' excluding solvent and model-map (green), are plotted across resolution. Map and model validation were performed in Phenix 1.19.2.

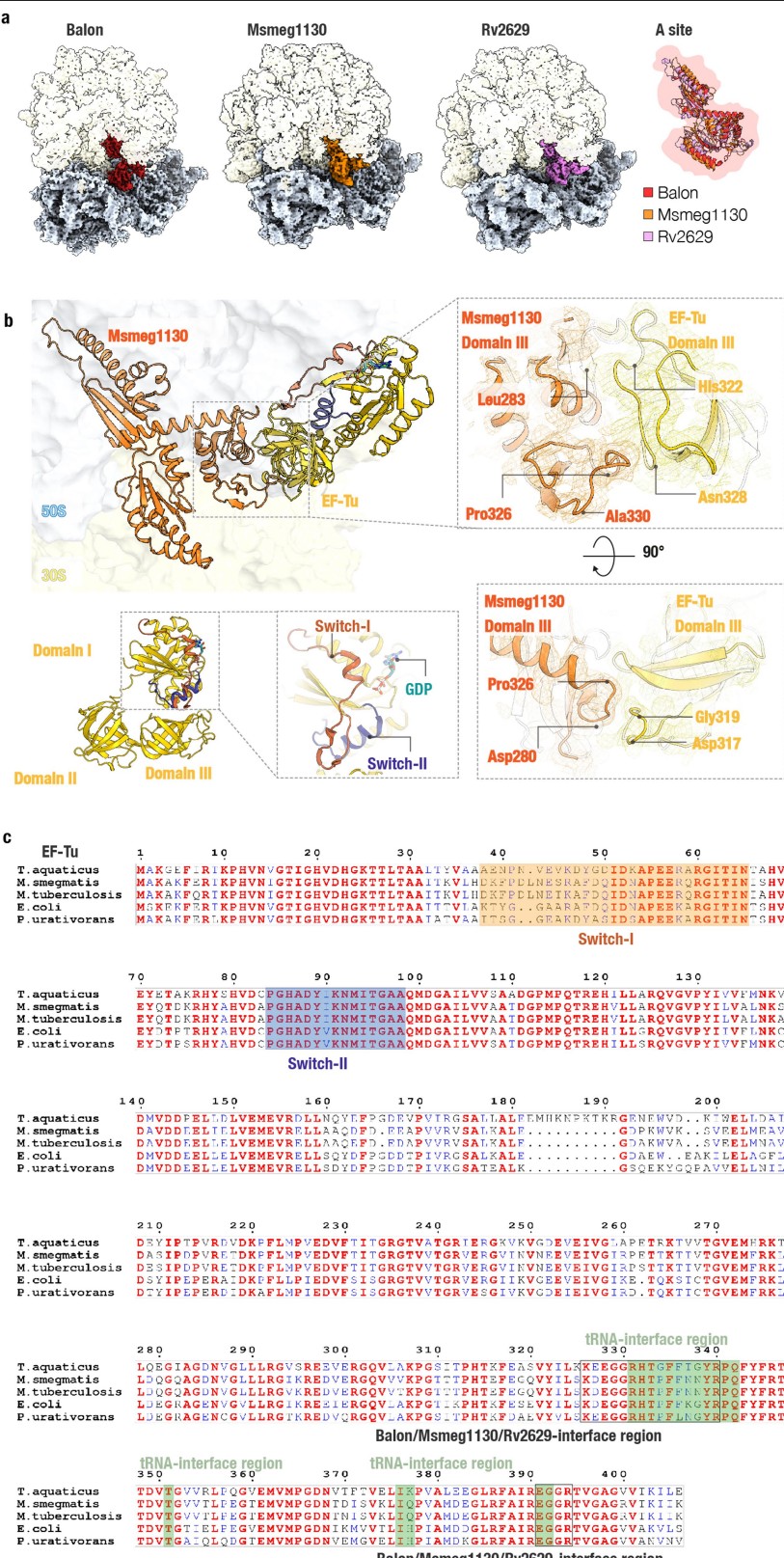

**Extended Data Fig. 8 | Comparison of the ribosomal and EF-Tu-binding sites of Balon and its homologues from different species shows the conservation of EF-Tu/Balon/ribosome association in different bacteria.** (**a**) Cryo-EM maps comparing the binding of Balon and its homologues Msmeg1130 (from *M. smegmatis*) and Rv2629 (*M. tuberculosis*) to the A site of bacterial ribosomes. (**b**) Zoomed-in view illustrating the protein/protein interaction interface for the ribosome-bound complex of Msmeg1130/EF-Tu.

(**c**) Multiple sequence alignment for EF-Tu from *T. aquaticus*, *P. urativorans*, *M. smegmatis*, *M. tuberculosis*, and *E. coli*, illustrating the high conservation of the Balon-binding site in EF-Tu from different bacteria. This panel shows that Balon and its mycobacterial homologues Msmeg1130 and Rv2629 recognize the same residues in domain III of EF-Tu that are involved in tRNA-binding (highlighted in green).

**Extended Data Fig. 9 | Comparison of Balon-mediated ribosome hibernation with the mechanisms of canonical protein synthesis and canonical ribosome hibernation.** (**a**) During normal protein synthesis, the elongation factor EF-Tu(GTP) delivers substrates for protein synthesis to the ribosomal A site and rapidly dissociates from the ribosome after GTP hydrolysis. (**b**) During starvation and stress, ribosomes become vacant after completing a cycle of protein synthesis, before binding to hibernation factors. EF-Tu is not required for this. (**c**) As in normal protein synthesis, Balon-mediated ribosome hibernation also involves the elongation factor EF-Tu and may occur while ribosomes remain associated with mRNAs and peptidyl-tRNA. However, unlike normal protein synthesis, this EF-Tu involvement appears to be antagonized by GTP, allowing only the GDP-bound form of EF-Tu to associate with hibernating ribosomes.

## Extended Data Table 1 | Cryo-EM data collection, refinement and validation parameters

| | Structure 1 Pu70S/Balon/RaiA (EMBD-19067) (PDB 8RD8) | Structure 2 Pu70S/Balon/ tRNA/mRNA (EMBD-19076) (PDB 8RDV) | Structure 3 Pu70S/Balon/ EF-Tu(GDP) (EMBD-19077) (PDB 8RDW) | Structure 4 Ms70S/ Msmeg1130 (EMBD-43074) (PDB 8V9J) | Structure 5 Ms70S/ Rv2629 (EMBD-43075) (PDB 8V9K) | Structure 6 (Composite) Ms70S/ Msmeg1130/ EF-Tu(GDP) (EMBD-43076)* (PDB 8V9L) |
|---|---|---|---|---|---|---|
| **Data collection and processing** | | | | | | |
| Magnification | 165k | 165k | 165k | 105k | 96k | 105k |
| Voltage (kV) | 300 | 300 | 300 | 300 | 300 | 300 |
| Electron exposure (e–/Å$^2$) | 40 | 40 | 40 | 40 | 40 | 40 |
| Defocus range (μm) | −2.4 to −0.9 | −2.4 to −0.9 | −2.4 to −0.9 | −0.8 to −2.0 | −1 to −2.3 | −0.8 to −2.0 |
| Pixel size (Å) | 0.723 | 0.723 | 0.723 | 0.839 | 0.85 | 0.839 |
| Symmetry imposed | C1 | C1 | C1 | C1 | C1 | C1 |
| Final particle images (no.) | 53,372 | 30,469 | 12,151 | 302,411 | 147,778 | 101,909 |
| Map resolution (Å) | 2.62 | 2.60 | 2.74 | 3.1 | 3.1 | 3.0 |
| FSC threshold | 0.143 | 0.143 | 0.143 | 0.143 | 0.143 | 0.143 |
| Map resolution range (Å) | 2.39 – 37.28 | 2.42 – 33.45 | 2.50 – 8.09 | 1.8 – 45.5 | 1.8 – 42.0 | 1.8 – 49.0 |
| **Refinement** | | | | | | |
| Model resolution (Å) | 3.06 | 3.1 | 3.4 | 3.2 | 3.3 | 3.2 |
| FSC threshold | 0.5 | 0.5 | 0.5 | 0.5 | 0.5 | 0.5 |
| Map sharpening B factor (Å$^2$) | −44.95 | −40.01 | −30 | −64 | −58 | See* |
| Model composition | | | | | | |
| Non-hydrogen atoms | 141,221 | 142,064 | 144,733 | 153,055 | 153,252 | 155,647 |
| Protein residues | 6,211 | 6,097 | 6,331 | 6,551 | 6,559 | 6,849 |
| Ligands: Mg$^{2+}$ | 2 | 0 | 1 | 2 | 2 | 2 |
| B factors (Å$^2$) (min/max/mean) | | | | | | |
| Protein | 3.7/98.4/32.6 | 3.6/98.4/32.5 | 3.6/98.8/35.2 | 16.3/176.9/77.9 | 18.0/165.0/75.9 | 13.8/120.6/52.4 |
| Nucleotide | 0.8/97.9/30.5 | 0.8/163.2/31.3 | 0.8/99.2/31.2 | 18.3/260.6/74.6 | 19.8/253.2/75.8 | 4.1/174.6/60.5 |
| Ligand | 30.0/30.0/30.0 | – | 30.0/30.0/30.0 | 55.3/59.5/57.4 | 52.7/55.3/54.0 | 49.9/53.9/51.9 |
| R.m.s. deviations | | | | | | |
| Bond lengths (Å) | 0.007 | 0.006 | 0.009 | 0.003 | 0.003 | 0.005 |
| Bond angles (°) | 0.744 | 0.695 | 0.766 | 0.556 | 0.541 | 0.610 |
| Validation | | | | | | |
| MolProbity score | 2.35 | 2.17 | 1.73 | 1.61 | 1.71 | 1.68 |
| Clashscore | 7.57 | 7.75 | 6.79 | 5.52 | 7.40 | 6.54 |
| Poor rotamers (%) | 7.12 | 4.08 | 0.13 | 0 | 0 | 0 |
| Ramachandran plot | | | | | | |
| Favoured (%) | 95.81 | 95.88 | 94.83 | 95.59 | 95.64 | 95.34 |
| Allowed (%) | 4.03 | 4.06 | 5.07 | 4.38 | 4.33 | 4.59 |
| Disallowed (%) | 0.16 | 0.07 | 0.11 | 0.03 | 0.03 | 0.07 |

The table shows the statistics for the complexes of 70S ribosomes from *P. urativorans* (*Pu*70S) and *M. smegmatis* (*Ms*70S).

Chris H. Hill
Matthieu G. Gagnon

# Reporting Summary

## Statistics

For all statistical analyses, confirm that the following items are present in the figure legend, table legend, main text, or Methods section.

| n/a | Confirmed | |
|---|---|---|
| ☐ | ☒ | The exact sample size (*n*) for each experimental group/condition, given as a discrete number and unit of measurement |
| ☐ | ☒ | A statement on whether measurements were taken from distinct samples or whether the same sample was measured repeatedly |
| ☒ | ☐ | The statistical test(s) used AND whether they are one- or two-sided *Only common tests should be described solely by name; describe more complex techniques in the Methods section.* |
| ☒ | ☐ | A description of all covariates tested |
| ☒ | ☐ | A description of any assumptions or corrections, such as tests of normality and adjustment for multiple comparisons |
| ☒ | ☐ | A full description of the statistical parameters including central tendency (e.g. means) or other basic estimates (e.g. regression coefficient) AND variation (e.g. standard deviation) or associated estimates of uncertainty (e.g. confidence intervals) |
| ☒ | ☐ | For null hypothesis testing, the test statistic (e.g. *F*, *t*, *r*) with confidence intervals, effect sizes, degrees of freedom and *P* value noted *Give P values as exact values whenever suitable.* |
| ☒ | ☐ | For Bayesian analysis, information on the choice of priors and Markov chain Monte Carlo settings |
| ☒ | ☐ | For hierarchical and complex designs, identification of the appropriate level for tests and full reporting of outcomes |
| ☒ | ☐ | Estimates of effect sizes (e.g. Cohen's *d*, Pearson's *r*), indicating how they were calculated |

*Our web collection on statistics for biologists contains articles on many of the points above.*

## Software and code

Policy information about availability of computer code

| Data collection | The cryo-EM data were collected using the Smart EPU Software for single particle analysis by Thermofisher Scientific. The bacterial growth curves were recordered using the Agilent BioTek Gen 5 software package. |
|---|---|
| Data analysis | The models were created using Coot v0.8.9.2<br>The models were analyzed and visualized using ChimeraX v1.4 and Chimera v1.14<br>The cryo-EM snapshots were processed using Relion v3.1 and CryoSPARC v4.3.0<br>The models were refined using Phenix v1.20.1-4487 |

For manuscripts utilizing custom algorithms or software that are central to the research but not yet described in published literature, software must be made available to editors and reviewers. We strongly encourage code deposition in a community repository (e.g. GitHub). See the Nature Portfolio guidelines for submitting code & software for further information.

## Data

Policy information about availability of data

All manuscripts must include a data availability statement. This statement should provide the following information, where applicable:
- Accession codes, unique identifiers, or web links for publicly available datasets
- A description of any restrictions on data availability
- For clinical datasets or third party data, please ensure that the statement adheres to our policy

DATA AVAILABILITY
The Supplementary Data 1-9 presented in this manuscript are freely accessible and available through the public repository of data, FigShare, using the following link: https://figshare.com/s/374a95769c5f7e9cdc04. The PDB accession codes for Structures 1-6, along with the associated cryo-EM maps will be made available before the publication of this manuscript via the Protein Data Bank. Currently, these structural data are shared through a freely accessible link on Google Drive.

## Research involving human participants, their data, or biological material

Policy information about studies with human participants or human data. See also policy information about sex, gender (identity/presentation), and sexual orientation and race, ethnicity and racism.

| | |
|---|---|
| Reporting on sex and gender | We did not study any human populations in our study. |
| Reporting on race, ethnicity, or other socially relevant groupings | We did not study any human populations in our study. |
| Population characteristics | We did not study any human populations in our study. |
| Recruitment | We did not study any human populations in our study. |
| Ethics oversight | We did not study any human populations in our study. |

Note that full information on the approval of the study protocol must also be provided in the manuscript.

# Field-specific reporting

Please select the one below that is the best fit for your research. If you are not sure, read the appropriate sections before making your selection.

☒ Life sciences          ☐ Behavioural & social sciences          ☐ Ecological, evolutionary & environmental sciences

For a reference copy of the document with all sections, see nature.com/documents/nr-reporting-summary-flat.pdf

# Life sciences study design

All studies must disclose on these points even when the disclosure is negative.

| | |
|---|---|
| Sample size | Each cryo-EM map presented in this study was calculated by using at least 10,287 individual ribosome snapshots to reconstruct the three-dimensional volume of the ribosome. |
| Data exclusions | Some ribosome images were excluded from the analysis based on the signal corresponding to the ribosomal ligands in the masked regions as described in detail in the Materials and Methods and Supplementary Figure 2. |
| Replication | To measure the impact of deletions of the Balon-coding gene on P. arcticus growth, each measurement was made eight times using different cell samples for each measurement. |
| Randomization | N/A |
| Blinding | N/A |

# Reporting for specific materials, systems and methods

We require information from authors about some types of materials, experimental systems and methods used in many studies. Here, indicate whether each material, system or method listed is relevant to your study. If you are not sure if a list item applies to your research, read the appropriate section before selecting a response.

## Materials & experimental systems

| n/a | Involved in the study |
|---|---|
| ☒ ☐ | Antibodies |
| ☒ ☐ | Eukaryotic cell lines |
| ☒ ☐ | Palaeontology and archaeology |
| ☒ ☐ | Animals and other organisms |
| ☒ ☐ | Clinical data |
| ☒ ☐ | Dual use research of concern |
| ☒ ☐ | Plants |

## Methods

| n/a | Involved in the study |
|---|---|
| ☒ ☐ | ChIP-seq |
| ☒ ☐ | Flow cytometry |
| ☒ ☐ | MRI-based neuroimaging |

## Plants

| Seed stocks | We did not study plants in our study. |
|---|---|
| Novel plant genotypes | We did not study plants in our study. |
| Authentication | We did not study plants in our study. |

