## [Peer Review File · Nature]

Manuscript Title: A new family of bacterial ribosome hibernation factors

Reviewer Comments & Author Rebuttals

Reviewer Reports on the Initial Version:

Referees' comments:

Referee #1 (Remarks to the Author):

In the present manuscript Helena-Bueno et al describe in detail a new bacterial ribosome inactivating protein identified from the cold-adapted bacterium *P. urativorans*. The new factor, termed Balon, was found to bind ribosomes at two distinct states, one together with the known hibernating factor RaiA and in a second ribosomal state with canonical tRNA in a P/P state and mRNA. This second state is also associated with the universally conserved translational GTPase EF-Tu. A subsequent extensive bioinformatic study revealed a widespread distribution of Balon-like proteins in multiple bacterial species probably underlying an important mechanism for ribosome protection mediated by this family of proteins.

The study is well designed and properly executed, revealing a new bacterial ribosomal factor that resembles eukaryotic translation factors (specifically eRF1) what may point towards a common origin or perhaps converging evolution between eukaryotes and prokaryotes at least in this specific aspect of translation.

-Major concern:

This reviewer finds a strong weakness in the study, and this has to do with the identification and characterization of the role of EF-Tu in binding and delivering balon to the ribosome. The third cryoEM map presented by the authors claiming better density for EF-Tu is still very poor in the region occupied by EF-Tu: there is barely side chain information and the post-processed maps showed exhibit poor quality and even discontinuous density. With these deficiencies in map quality around the GTPase binding site of the ribosome is nearly impossible to identify the nucleotide state of EF-Tu: is EF-Tu bound to the ribosome in a GTP state? post-hydrolysis? May be in a GDP bound state? The role and functional cycle of EF-Tu in aa-tRNA delivery to the ribosome is well known: a ternary (GTP/EF-Tu/aa-tRNA) complex which is a biochemically stable entity, binds and samples the ribosomal A-site in search for a complementary mRNA codon. If the codon-anticodon interaction is validated by the ribosomal A-site, GTP hydrolysis is induced by the displacement of a key histidine residue in EF-Tu by the sarcin-ricin loop, a component of the large ribosomal subunit. Once EF-Tu is in its GDP state, a conformational change in the protein decreases its affinity for the ribosome liberating the aa-tRNA to accommodate its CCA-AA-end near the peptidyl ribosomal site while the EF-Tu/GDP complex leaves the ribosomal A site. Does EF-Tu play a similar role in the delivery of balon to the ribosomal A site? Is the balon/EF-Tu/GTP complex a biochemically stable entity that can be purified and characterized? Which are the nucleotide requirements of EF-Tu for an efficient delivery of balon to the ribosomes? Does it matter the presence of a P site tRNA or mRNA for the efficient delivery of balon under stress conditions? All these questions, in the opinion of this reviewer should be addressed.

-Minor concerns:

The authors should provide a better graphical display of the local resolution values for the ribosome ligands. Supplementary Figure S4 only show far views of maps colored according to local resolution values what makes virtually impossible to judge the cryoEM density quality for the ribosomal ligands specially EF-Tu. A zoomed view centered around the functional sites of the ribosome with a clear view of the density for the factors colored according to local resolution calculations is necessary.

The refinement statistics of the atomic model for the three maps are not ideal. The reported molprobity clashcores (11.25, 14.79 and 13.58) are too high for this resolution range. The authors should improve these numbers probably by introducing a second refining step with refmac after real space refinement in phenix.

The manuscript will benefit from less hyperbolic expressions of the type:

"These findings call for a revision of our model of bacterial translation inferred from common model organisms and hold numerous implications for how we understand and study ribosome dormancy. "

"Finally, we found that Balon has an anomalously high rate of sequence evolution."

"providing a key missing link in our understanding of ribosome protection during translational arrest."

"Importantly, our discovery of Balon revises the model of ribosome dormancy and argues against a single generalized mechanism inferred from studies in *E. coli* "

"binding to Balon is significant because EF-Tu is the target of several antibiotics (e.g. GE2270A and kirromycin) and understanding its mechanism of action serves as the basis for developing safer and more effective clinical drugs"

Finally, as a Spanish speaking person I must notice that the right spelling of balon in Spanish is balón, which translates as ball or football in English as the authors rightly noticed. Balon, without the accent in the last syllable does not exist in the Spanish language.

Referee #2 (Remarks to the Author):

Helena-Bueno and co-authors report a discovery of a new bacterial protein that binds ribosomes in a *Psychrobacter* species. Cryo-EM analyses reveal the protein in the ribosomal A site, bound to both the 30S and 50S subunits. In addition, the protein interacts with EF-Tu, whose normal function is to deliver aminoacyl-tRNA to the ribosome during elongation. The protein's fold is similar to that of archaeal/eukaryotic release factors, which is unprecedented in bacteria. Yet the protein does not have the features of the release factor to terminate translation or of the similarly folded eukaryotic stress-response factor Pelota to ribosome rescue by tRNA and subunit dissociation. The authors find homologous sequences in 23 of 27 major bacterial phyla, highlighting their functional importance. The discovery of the new intriguing bacterial factor, termed Balon, is therefore exciting and significant.

The authors propose that Balon is a hibernation factor that protects the ribosomal functional centers on the 30S and 50S subunits from stress-induced degradation. In this capacity, Balon's function would be similar to that of other hibernation factors, such as Hpf/RaiA binding to the mRNA tunnel. While this and some other hypotheses put forth by the authors are compelling, the study suffers from the lack of experiments testing these ideas. In fact, some data argue against the main hypothesis. For example, Balon levels in the bacterium are similar at different conditions (with/without cold shock), arguing that the protein may have a function during normal bacterial growth. The protein may participate in ribosome quality control or in ribosome biogenesis – these functions must be tested by cellular and/or biochemical experiments.

The authors' evolutionary analyses yield important findings, however some interpretations are misleading. For example, the authors make a strong point about Balon being a "missing link" and "the first experimental evidence of the existence of eukaryote-type translation factors in bacterial cells". But numerous translation factors are conserved between bacteria and eukaryotes (IF2/eIF5B, EF-Tu/eEF1A, EF-G/eEF2, to name a few), making such claims misleading. In sum, this work must be substantially revised prior to publication.

Criticisms:

1. The major criticism is that the "hibernation" function of Balon is not shown. Furthermore, other

possible functions have not been tested.

2. In the figures, only the secondary structure is shown for most regions where Balon interacts with the ribosome. But the reported resolution appears to allow side-chain assignments, which define the specific Balon binding. Show density maps and side chains for these regions:

- a. "Lasso" binding near A2602 (Fig. 2B)
- b. HP motif interacting with the decoding center (Fig. 2B)
- c. How is bL27 recognized by the "bL27 trap" side chains? (Fig. 2B and 5B)

3. The discovery of EF-Tu interaction with Balon necessitates a deeper discussion of EF-Tu, which is not known to deliver molecules other than aa-tRNA:

- a. Why is EF-Tu bound, in contrast to aa-tRNA complexes, from which EF-Tu quickly dissociates upon delivery to the ribosome?
- b. What's the GTPase center structural and functional state? (GTP-, GDP-bound?) What are the conformations of the switch loops? How close is it to the SRL?
- c. How does EF-Tu bind/recognize Balon and how does this recognition differ from that of aa-tRNA?
- d. Does *Psychrobacter* EF-Tu (sequence/structure) differ from that in bacteria that do not express Balon? If there are differences, do they suggest that EF-Tu in *Psychrobacter* has diverged to interact with Balon?

4. The "missing link" and "the first experimental evidence of the existence of eukaryote-type translation factors in bacterial cells" claims are misleading and should be clarified or removed.

5. In evolutionary considerations, the authors propose that the direction of eRF1-like gene transfer is "from archaea or eukaryotes to bacteria". Why not from bacteria to archaea/ eukaryotes?

6. Have the authors analyzed eukaryotic genomes in search of Balon-like mitochondrial factors?

Minor comments:

1. In Fig. 1B, "Rai1A" label points to 30S, not to Rai1A in the figure.
2. On page 9, the authors refer to "Supplementary Data S1". Should this be "Supplementary Figure S1"?
3. On page 9, "...bound to heterologous mRNA and peptidyl-tRNA..." – why heterologous (from another organism)? Did they mean "heterogeneous"?
4. On page 10, "in the number of particles with bound EF-Tu (Fig. S5," – but no EF-Tu is shown in Fig. S5.
5. Change the title of the section "The catalytic site of..." to "The peptidyl transferase center of...".
6. The central domain of eRF1 is termed the "M" domain in the field. It is best to stick to this nomenclature in the text and figures to avoid confusion.

Referee #3 (Remarks to the Author):

The study by Helena-Bueno et al. describes the structural features of a newly discovered translation hibernation factor, termed "Balon". Balon was discovered to bind ribosomes during cold exposure in the psychrophilic bacterium *P. urativorans*. The authors show structural evidence that Balon binds ribosomes, how the protein may inactivate ribosomes and its conformational -but not functional - relationship to archeal RF1. Interestingly, the protein seems present in about 20% of the investigated prokaryotic species and was obviously overlooked before.

The manuscript contains a very nice collection of high-quality structural data that gives beautiful insights into the mechanism of hibernation. The figures are very nicely illustrated, the methods are comprehensive and the text is overall well written.

However, I am not fully convinced that the observed mechanism has really such revolutionary

character that requires a "revision of our model of bacterial translation" as stressed in the abstract and throughout the text. Clearly, literature was very focused on the mechanistic understanding of few hibernation factors in the prominent model systems. Thus, the precise function of other hibernation factors – not just in prokaryotes – still awaits mechanistic characterization. However, the protein clearly exhibits a "classical" hibernation function and I do not see Balon's function being so surprisingly different as claimed. In that context, it would have been nice to see the physiological consequences of the absence of Balon during stress response or dormancy. Such an assay would have been essential to dissect if Balon has really such unique features qualifying it to be distinct from other hibernation factors.

Besides that, I do only have minor comments concerning specific sections of the text.

Abstract:

The second sentence is a bit misleading, it sounds as if hibernation factors damage ribosomes. Please rephrase.

Introduction:

Fourth paragraph: the authors used the term hibernation factors before and suddenly switch to the term "dormancy factor" this is confusing. Please stay consistent with one term.

Results:

Overall, the results section reads a bit superficial and could benefit from a deeper explanation of the finding.

The authors use eRF1, aeRF1, aRF1 throughout the text and figures. Is there a better way to distinguish between these forms? Non-experts might be easily confused by these terms. Also better explain all abbreviations in the legends (e.g. "Ce").

Page 10, second paragraph. I am not sure if Balon should be classified as bona fide translation factor. It is clearly a hibernation factor, which adopts a conformation as other translation factors. But it exhibits no translation factor activity.

Figure 5A and B:

It might be interesting to show the involved residues of Balon in the cartoon to illustrate its function in inactivating the PTC.

Author Rebuttals to Initial Comments:

Referees' comments:

Referee #1 (Remarks to the Author):

In the present manuscript Helena-Bueno et al describe in detail a new bacterial ribosome inactivating protein identified from the cold-adapted bacterium *P. urativorans*. The new factor, termed Balon, was found to bind ribosomes at two distinct states, one together with the known hibernating factor RaiA and in a second ribosomal state with canonical tRNA in a P/P state and mRNA. This second state is also associated with the universally conserved translational GTPase EF-Tu. A subsequent extensive bioinformatic study revealed a widespread distribution of Balon-like proteins in multiple bacterial species probably underlying an important mechanism for ribosome protection mediated by this family of proteins.

The study is well designed and properly executed, revealing a new bacterial ribosomal factor that resembles eukaryotic translation factors (specifically eRF1) what may point towards a common origin or perhaps converging evolution between eukaryotes and prokaryotes at least in this specific aspect of translation.

We thank the reviewer for this positive appraisal of our work.

-Major concern:

This reviewer finds a strong weakness in the study, and this has to do with the identification and characterization of the role of EF-Tu in binding and delivering Balon to the ribosome. The third cryoEM map presented by the authors claiming better density for EF-Tu is still very poor in the region occupied by EF-Tu: there is barely side chain information and the post-processed maps showed exhibit poor quality and even discontinue density.

We have addressed this concern by recollecting the entire dataset at 300 kV and reprocessing the images, resulting in higher resolution maps for all structures. Additionally, we performed focused classification with signal subtraction on i) the entire EF-Tu molecule and ii) EF-Tu domain I (for which we succeeded in separating two subtly different orientations, tilted by $\sim 10^\circ$). Although local resolution for EF-Tu is still limited to 2.5 - 4.5Å, this has improved the density for EF-Tu (see revised map 3; **Fig. S5, S6**). To further aid interpretation and to avoid artefacts of over-sharpening, we also provide postprocessed maps filtered to 4Å and 6Å.

With these deficiencies in map quality around the GTPase binding site of the ribosome is nearly impossible to identify the nucleotide state of EF-Tu: is EF-Tu bound to the ribosome in a GTP state? post-hydrolysis? May be in a GDP bound state?

Using this new cryo-EM map, we show that EF-Tu likely adopts the GDP-bound conformation, as evidenced by the overall conformation of the nucleotide-binding domain, the position of switch loops, and density in the nucleotide-binding pocket consistent with the presence of GDP or GDP/Mg²⁺. This new information is now shown in **Figure 5** and **Figure S6**.

The role and functional cycle of EF-Tu in aa-tRNA delivery to the ribosome is well known: a ternary (GTP/EF-Tu/aa-tRNA) complex which is a biochemically stable entity, binds and samples the ribosomal A-site in search for a complementary mRNA codon. If the codon-anticodon interaction is validated by the ribosomal A-site, GTP hydrolysis is induced by the displacement of a key histidine residue in EF-Tu by the sarcin-ricin loop, a component of the large ribosomal subunit. Once EF-Tu is

in its GDP state, a conformational change in the protein decreases its affinity for the ribosome liberating the aa-tRNA to accommodate its CCA-AA-end near the peptidyl ribosomal site while the EF-Tu/GDP complex leaves the ribosomal A site. Does EF-Tu play a similar role in the delivery of Balon to the ribosomal A site?

Is the Balon/EF-Tu/GTP complex a biochemically stable entity that can be purified and characterized? Which are the nucleotide requirements of EF-Tu for an efficient delivery of balon to the ribosomes? Does it matter the presence of a P site tRNA or mRNA for the efficient delivery of Balon under stress conditions? All these questions, in the opinion of this reviewer should be addressed.

These are very interesting questions, which we have addressed through several additional experiments. We have also re-written the corresponding sections of the results and discussion. Briefly:

- 1) Our new cryo-EM map (described in our response to the previous point by Reviewer 1) shows that EF-Tu exists in the GDP-bound state when in complex with ribosomes and Balon.
- 2) We performed a comparative analysis EF-Tu's structural recognition by the C-terminal domains of Balon, aeRF1, and Pelota, as well as aminoacyl-tRNAs. We find that, while Balon, aeRF1, and Pelota all use their C-terminal domain to associate with EF-Tu, Balon employs a distinct structural mechanism involving a unique β -loop not present in aeRF1/Pelota that shifts the EF-Tu binding surface ~ 20 Å away. We show that this altered binding surface makes Balon incapable of forming a stable complex with the GTP-bound EF-Tu.
- 3) Using cryo-EM analysis of *in vitro* assembled ribosome complexes, we demonstrate stable complex formation between Balon and EF-Tu only when the ribosome, Balon, EF-Tu, and GDP are all present in the reaction. Using size-exclusion chromatography to test EF-Tu/Balon complex formation, we were unable to detect Balon:EF-Tu interaction in the presence of GTP (the non-hydrolyzable analog GDPCP) – in stark contrast to aeRF1 and Pelota. Furthermore, using cryo-EM to reconstruct the complex between the ribosome, Balon and EF-Tu, we observed that GDPCP inhibited the association of EF-Tu with the ribosome-bound Balon.

Taken together, our findings suggest a unique mechanism for EF-Tu association with Balon that does not mirror the recruitment mechanism of aeRF1 and Pelota. Our data imply that this mechanism involves either (1) Balon association with the ribosomal A site and a subsequent recruitment of EF-Tu(GDP) or (2) Balon recruitment by EF-Tu(GDP) via the weak interactions between EF-Tu, Balon and the ribosomal L7/12 stalk. In either of these scenarios, Balon—unlike aminoacyl-tRNAs, aeRF1, and Pelota—does not engage with the GTP-bound form of EF-Tu, providing a possible explanation for why Balon does not interfere with protein synthesis during normal growth conditions, when cells contain abundant levels of GTP and a high GTP/GDP ratio. Therefore, by contrast to aminoacyl-tRNAs, aeRF1 and Pelota, Balon loading in the A site appears to bypass not only the step of mRNA codon verification but also the step GTP hydrolysis, explaining how Balon is able to bind to nearly all cellular ribosomes during starvation and stress). Importantly, our work provides the first evidence for EF-Tu's role in ribosome hibernation, which will require numerous future studies to fully elucidate this role.

-Minor concerns:

The authors should provide a better graphical display of the local resolution values for the ribosome ligands. Supplementary Figure S4 only show far views of maps colored according to local resolution values what makes virtually impossible to judge the cryoEM density quality for the ribosomal ligands specially EF-Tu. A zoomed view centered around the functional sites of the ribosome with a clear view of the density for the factors colored according to local resolution calculations is necessary.

Thank you very much for this comment. These images are now shown in Figure S5.

The refinement statistics of the atomic model for the three maps are not ideal. The reported molprobity clashcores (11.25, 14.79 and 13.58) are too high for this resolution range. The authors should improve these numbers probably by introducing a second refining step with refmac after real space refinement in phenix.

We thank the reviewer for this comment. Upon further investigation, most of these clashes were occurring during refinement in poor quality density for flexible proteins (e.g. L7/L12 stalk). We have decided that these flexible regions do not merit the building of side-chains, and have truncated them. This has improved the clash score to 6.15, 6.83 and 6.79, and made it clear that they should not be interpreted at the same level as the better resolved areas of the map.

The manuscript will benefit from less hyperbolic expressions of the type: “These findings call for a revision of our model of bacterial translation inferred from common model organisms and hold numerous implications for how we understand and study ribosome dormancy.” “Finally, we found that Balon has an anomalously high rate of sequence evolution.” “providing a key missing link in our understanding of ribosome protection during translational arrest.” “Importantly, our discovery of Balon revises the model of ribosome dormancy and argues against a single generalized mechanism inferred from studies in E. coli “ “binding to Balon is significant because EF-Tu is the target of several antibiotics (e.g. GE2270A and kirromycin) and understanding its mechanism of action serves as the basis for developing safer and more effective clinical drugs”

We have toned down this type of language throughout the manuscript.

Finally, as a Spanish speaking person I must notice that the right spelling of balon in Spanish is balón, which translates as ball or football in English as the authors rightly noticed. Balon, without the accent in the last syllable does not exist in the Spanish language.

Thank you for this comment – we completely agree. However, unfortunately it will generally not be possible to carry the accent forward into various databases where the gene and protein nomenclature will need to be annotated. Therefore, we have chosen instead to introduce the nomenclature as the following: “We therefore name this protein Balon (after “balón”, Spanish for “ball”) to highlight its distant structural similarity to Pelota (also Spanish for “ball”).” We hope that this addresses the reviewer’s concern.

Referee #2 (Remarks to the Author):

Helena-Bueno and co-authors report a discovery of a new bacterial protein that binds ribosomes in a Psychrobacter species. Cryo-EM analyses reveal the protein in the ribosomal A site, bound to both the 30S and 50S subunits. In addition, the protein interacts with EF-Tu, whose normal function is to deliver aminoacyl-tRNA to the ribosome during elongation. The protein’s fold is similar to that of archaeal/eukaryotic release factors, which is unprecedented in bacteria. Yet the protein does not

have the features of the release factor to terminate translation or of the similarly folded eukaryotic stress-response factor Pelota to ribosome rescue by tRNA and subunit dissociation. The authors find homologous sequences in 23 of 27 major bacterial phyla, highlighting their functional importance. The discovery of the new intriguing bacterial factor, termed Balon, is therefore exciting and significant.

We thank the reviewer for their enthusiasm for the subject matter.

The authors propose that Balon is a hibernation factor that protects the ribosomal functional centers on the 30S and 50S subunits from stress-induced degradation. In this capacity, Balon's function would be similar to that of other hibernation factors, such as Hpf/RaiA binding to the mRNA tunnel. While this and some other hypotheses put forth by the authors are compelling, the study suffers from the lack of experiments testing these ideas. In fact, some data argue against the main hypothesis. For example, Balon levels in the bacterium are similar at different conditions (with/without cold shock), arguing that the protein may have a function during normal bacterial growth. The protein may participate in ribosome quality control or in ribosome biogenesis – these functions must be tested by cellular and/or biochemical experiments.

The authors' evolutionary analyses yield important findings, however some interpretations are misleading. For example, the authors make a strong point about Balon being a “missing link” and “the first experimental evidence of the existence of eukaryote-type translation factors in bacterial cells”. But numerous translation factors are conserved between bacteria and eukaryotes (IF2/eIF5B, EF-Tu/eEF1A, EF-G/eEF2, to name a few), making such claims misleading. In sum, this work must be substantially revised prior to publication.

Criticisms:

1. The major criticism is that the “hibernation” function of Balon is not shown. Furthermore, other possible functions have not been tested.

Thank you for this important comment. To address this concern, we have added the following data supporting the idea that Balon is a *bona fide* hibernation factor:

- 1) Using an *in vitro* translation assay, we show biochemically that Balon homologs effectively inhibit protein synthesis to similar levels as the previously described hibernation factor RaiA: a 15-fold reduction in the rate of protein synthesis.
- 2) We purified native ribosomes from a long-term stationary culture of *P. urativorans* cells, and analysed them by cryo-EM. This showed that Balon binds to practically all cellular ribosomes in these conditions – demonstrating a more generalized response to stress conditions beyond cold-shock.
- 3) We investigated this genetically, revealing that knockout of Balon impaired acute cellular recovery from stress conditions.

To address this concern further, we have corrected our discussion to state that we cannot exclude the possibility that, as a previously uncharacterized cellular protein, Balon may play other functions besides hibernation in protein synthesis or even in a cell, and further studies are needed to understand the biology of this protein.

2. In the figures, only the secondary structure is shown for most regions where Balon interacts with the ribosome. But the reported resolution appears to allow side-chain assignments, which define the specific Balon binding. Show density maps and side chains for these regions:

- a. “Lasso” binding near A2602 (Fig. 2B)
- b. HP motif interacting with the decoding center (Fig. 2B)
- c. How is bL27 recognized by the “bL27 trap” side chains? (Fig. 2B and 5B)

We now show these maps and side chains in Fig. 3B of the revised manuscript.

3. The discovery of EF-Tu interaction with Balon necessitates a deeper discussion of EF-Tu, which is not known to deliver molecules other than aa-tRNA:

Thank you for this point – we have conducted several new experiments and analyses to address the role of EF-Tu, detailed in our response to Reviewer 1 above. We also specifically answer Reviewer 2’s questions below:

- a. **Why is EF-Tu bound, in contrast to aa-tRNA complexes, from which EF-Tu quickly dissociates upon delivery to the ribosome?**

Our revised Figure 5 illustrates that, in contrast to aeRF1 and Pelota, Balon uses a dissimilar mechanism of EF-Tu recognition that may explain its unusual ability to associate with the GDP-bound rather than GTP-bound form of EF-Tu and remain bound to EF-Tu(GDP) while also being fully accommodated by the ribosomal A site. We also show that, in addition to association with Balon, EF-Tu(GDP) forms multiple contacts with the ribosome: its domain I remains attached to the L7/L12-stalk of the ribosome, its domain II forms a previously observed contacts with 16S rRNA helix h5 in the 30S subunit, and its domain III binds not only to Balon but also to the tip of the sarcin-ricin loop of the ribosome (Fig. 5b). These contacts, including the previously unknown interaction between domain III and the tip of the sarcin-ricin loop, possibly explain a more stable association between EF-Tu(GDP) and Balon-bound ribosomes compared to the ribosomes bound to A-site tRNA in which domain III remains detached from the sarcin-ricin loop of the ribosome (Fig. S6).

- b. **What’s the GTPase center structural and functional state? (GTP-, GDP-bound?) What are the conformations of the switch loops? How close is it to the SRL?**

We provide higher-resolution cryo-EM maps and biochemical analyses to address this point. Our new cryo-EM dataset shows that the EF-Tu molecule has the GDP-bound conformation, as evident from the overall shape of the protein, the conformation of both switches, and the density in the nucleotide-binding pocket, which is consistent with the presence of GDP or GDP/Mg²⁺ (Fig. 5a-d). We also show by co-precipitation of purified proteins that Balon and EF-Tu can only form a stable complex in the presence of the ribosome and GDP – and not in the presence of GTP (Fig. 5f,g). In this conformation, the nucleotide-binding pocket of EF-Tu is ~28-29 Å away from the SRL in both EF-Tu conformations, as measured by the distance between the C α atom of His84 in EF-Tu (*E. coli* numbering) and the OP2 atom of 23S A2662 (*E. coli* numbering), as shown in Fig. S6.

- c. **How does EF-Tu bind/recognize Balon and how does this recognition differ from that of aa-tRNA?**

Our revised Figure 5 compares this recognition not only with that of aa-tRNA but also with aeRF1 and Pelota, illustrating that Balon uses an entirely different surface of its C-terminal

domain, making Balon structurally incompatible with the GTP conformation of EF-Tu. This is now discussed in detail in the text.

- d. Does Psychrobacter EF-Tu (sequence/structure) differ from that in bacteria that do not express Balon? If there are differences, do they suggest that EF-Tu in Psychrobacter has diverged to interact with Balon?**

We now show this information in the Figure S25. Contrary to this, our Figure S25 shows that the Balon-binding site in the EF-Tu molecule is conserved across species, and our biochemical and cryo-EM data on the Mycobacterial homolog of Balon, Msmeg1130, show that the EF-Tu/Balon interaction is likely conserved in other bacteria instead of being a unique property of EF-Tu/Balon from *Psychrobacter* species.

- 4. The “missing link” and “the first experimental evidence of the existence of eukaryote-type translation factors in bacterial cells” claims are misleading and should be clarified or removed.**

We apologise and agree that these claims could cause confusion. We have removed this section of the text entirely.

- 5. In evolutionary considerations, the authors propose that the direction of eRF1-like gene transfer is “from archaea or eukaryotes to bacteria”. Why not from bacteria to archaea/ eukaryotes?**

We agree with the reviewer that the answer to this question requires further investigation and have therefore removed our speculation that Balon-coding genes are the result of horizontal gene transfer from archaea or eukaryotes to bacteria. We will address this question in future studies.

- 6. Have the authors analyzed eukaryotic genomes in search of Balon-like mitochondrial factors?**

Yes, but our preliminary searches using HHMER or FoldSeek-based homology search revealed that cytosolic eRF1 factors and Pelota proteins are only apparent structural homologs of Balon in eukaryotic cells. We have not observed any apparent Balon homologs in mitochondria. We have therefore elected not to add this negative data to the manuscript – but remain open to the possibility of adding it to our supplemental data if reviewers and the editor feels strongly that this should be included.

Minor comments:

- 1. In Fig. 1B, “Rai1A” label points to 30S, not to Rai1A in the figure.**
- 2. On page 9, the authors refer to “Supplementary Data S1”. Should this be “Supplementary Figure S1”?**
- 3. On page 9, “...bound to heterologous mRNA and peptidyl-tRNA...” – why heterologous (from another organism)? Did they mean “heterogeneous”?**
- 4. On page 10, “in the number of particles with bound EF-Tu (Fig. S5,” – but no EF-Tu is shown in Fig. S5.**
- 5. Change the title of the section “The catalytic site of...” to “The peptidyl transferase center of...”.**
- 6. The central domain of eRF1 is termed the “M” domain in the field. It is best to stick to this nomenclature in the text and figures to avoid confusion.**

We thank the reviewer for these suggestions. All these corrections have been made in our rewritten manuscript.

Referee #3 (Remarks to the Author):

The study by Helena-Bueno et al. describes the structural features of a newly discovered translation hibernation factor, termed “Balon”. Balon was discovered to bind ribosomes during cold exposure in the psychrophilic bacterium *P. urativorans*. The authors show structural evidence that Balon binds ribosomes, how the protein may inactivate ribosomes and its conformational -but not functional - relationship to archeal RF1. Interestingly, the protein seems present in about 20% of the investigated prokaryotic species and was obviously overlooked before.

The manuscript contains a very nice collection of high-quality structural data that gives beautiful insights into the mechanism of hibernation. The figures are very nicely illustrated, the methods are comprehensive and the text is overall well written.

We thank the reviewer for their kind remarks on the quality of the writing.

However, I am not fully convinced that the observed mechanism has really such revolutionary character that requires a “revision of our model of bacterial translation” as stressed in the abstract and throughout the text. Clearly, literature was very focused on the mechanistic understanding of few hibernation factors in the prominent model systems. Thus, the precise function of other hibernation factors – not just in prokaryotes – still awaits mechanistic characterization. However, the protein clearly exhibits a “classical” hibernation function and I do not see Balon’s function being so surprisingly different as claimed. In that context, it would have been nice to see the physiological consequences of the absence Balon during stress response or dormancy. Such assay would have been essential to dissect if Balon has really such unique features qualifying it to be distinct from other hibernation factors.

These are great points that have helped us to strengthen the manuscript. Firstly, to address the reviewer’s concerns over the physiological relevance of Balon, we generated a knockout strain lacking Balon and showed that this deletion impairs recovery from dormancy (**Fig. 2g and Fig. S14b**) Secondly, we collected an additional ribosomal cryo-EM dataset in which, in contrast to the abrupt stress of cold-shock, cells were allowed to reach stationary phase and remain in dormancy for four days. Under these conditions, we find that Balon also binds to nearly all ribosomes, and, in contrast to the acute stress conditions, these stationary-phase ribosomes lack P-site tRNA (**Fig. 1c**). However, when cells are exposed to abrupt cold shock, Balon can bind to practically all cellular ribosomes even when a substantial fraction of them are still associated with peptidyl-tRNA. This sets Balon apart from other hibernation factors that can bind only vacant ribosomes.

We also wish to emphasize some unique features of Balon that we believe make it an important new addition to the ribosome hibernation factor landscape, expanding our definition of what a hibernation factor can be:

- 1) The ability of Balon to bind ribosomes concurrently with P-site tRNA and mRNA is unprecedented among previously described hibernation factors – and, as we explain in our discussion, could explain why Balon has not been discovered previously due to the common practice of filtering out ribosomes containing P-site tRNA in hibernation studies. We believe that this is an important point that will hopefully facilitate the discovery of additional hibernation factors in the future by revising this restrictive definition.

2) Balon binds to a unique site compared to previously described hibernation factors RaiA and RMF, expanding our view of how ribosomes can enter an inactive state.

3) The conserved involvement of EF-Tu in Balon's ribosome hibernation activity – while still requiring further study to fully understand – represents a new facet of the ribosomal stress response involving a previously unappreciated role for this essential factor.

4) Balon represents the first bacterial ribosome hibernation factor with homology to archaeoeukaryotic translation factors, providing insight into one possible evolutionary origin of hibernation factors in bacterial cells.

In our view, these points speak to the conceptual advance that the discovery of Balon provides to the ribosome hibernation field. However, we agree with the reviewer that claiming this revises our model of bacterial translation more broadly is likely an overstatement. We have revised the text accordingly to focus on these unique features and how they expand our current model of ribosome hibernation.

Besides that, I do only have minor comments concerning specific sections of the text.

Abstract:

The second sentence is a bit misleading, it sounds as if hibernation factors damage ribosomes. Please rephrase.

We have now rewritten the abstract to address this comment.

Introduction:

Fourth paragraph: the authors used the term hibernation factors before and suddenly switch to the term "dormancy factor" this is confusing. Please stay consistent with one term.

We now use "hibernation factor" for consistency.

Results:

Overall, the results section reads a bit superficial and could benefit from a deeper explanation of the finding.

We have extensively revised the entire Results section with this comment in mind and added substantial new data. We hope that this addresses the reviewer's concern, and we remain open to further textual revisions if requested.

The authors use eRF1, aeRF1, aRF1 throughout the text and figures. Is there a better way to distinguish between these forms? Non-experts might be easily confused by these terms. Also better explain all abbreviations in the legends (e.g. "Ce").

Thank you for pointing this out; we agree that this would help with clarity. We now use "aeRF1" to refer to the archaeo-eukaryotic release factor 1, and we have also corrected our figure legends to provide explanations for shorthands and acronyms.

Page 10, second paragraph. I am not sure if Balon should be classified as bona fide translation factor. It is clearly a hibernation factor, which adopts a conformation as other translation factors. But it exhibits no translation factor activity.

We agree, and we have corrected our text where necessary to refer to Balon as a hibernation factor, not a translation factor.

Figure 5A and B:

It might be interesting to show the involved residues of Balon in the cartoon to illustrate its function in inactivating the PTC.

We have now made these corrections in our revised **Fig. 3b**.

Reviewer Reports on the First Revision:

Referees' comments:

Referee #1 (Remarks to the Author):

The authors have addressed all my previous concerns, improving dramatically the quality of the cryoEM structures as well as the characterization of the role of EF-Tu. They have additionally included new structures from human pathogenic bacteria, expanding the impact of this work. I consider the manuscript ready for publication in its current form.

Referee #2 (Remarks to the Author):

The revised manuscript describing the discovery of ribosome-binding stress factor Balon has substantially improved upon revision with the addition of new cryo-EM analyses, and biochemical and cellular assays; my previous concerns are well-addressed. I recommend publication of this insightful manuscript. I have just a few minor suggestions that should be straightforward for the authors to implement.

Minor comments:

- On page 12: "...causing a fifteen-fold reduction in the rate of protein synthesis (Fig. 2h)".

Describe how the rates of protein synthesis were calculated.

- The short mention of Balon's overlap with antibiotic-binding sites and the corresponding panels 1g-I are confusing, in my opinion. The authors state that "...Balon's binding site overlaps with multiple ribosome-targeting antibiotics... (Fig. 1g-i). Thus, Balon ...occupies the ribosomal active sites, rendering them inaccessible to other molecules.". It is not clear how antibiotics are relevant in the context of normal ribosome activity, at least as described in the text. By contrast, these figure panels may imply that (1) Balon is involved in antibiotic resistance; or (2) that antibiotics prevent Balon binding to the ribosome. Neither of these scenarios are tested in this work. To avoid confusion, I suggest removing this discussion and figure panels, or moving them to Supplementary information.

- It is not clear from the density description which parts of the mRNA (in the P-tRNA-Balon structure) are resolved. Is the P-site codon resolved (despite heterogeneity, codon-anticodon base-pairing may have strong density; alternatively, this could be a specific codon – e.g. an AUG – which would suggest binding to initiating ribosomes)? How well is the mRNA codon/backbone resolved in the A site? mRNA tunnel? It would help readers if these findings are described clearly.

- Discussion could include an important aspect of stress response in bacteria that may underlie the functional role of Balon. Specifically, cryo-EM/biochemical data here show that Balon binds EF-Tu in the presence of GDP – rather than in the presence of GTP. This suggests that Balon could be a sensor of the GDP/GTP ratio in the cell. Under normal growth conditions, the GDP-bound state of EF-Tu is rare due to quick exchange of GDP with more abundant GTP molecules. By contrast, the ratio of GDP/GTP increases during stress. Moreover, GTP concentration is reduced due to its conversion to pppGpp and subsequently to ppGpp. It may be physiologically relevant that the latter binds to GTPases similarly to GDP; therefore, the cellular substrate for Balon could be EF-Tu*ppGpp. Given the already impressive body of work in this manuscript, it is not required for the authors to do experiments to test whether Balon binds EF-Tu with ppGpp; but if they decide to do so, they might find the missing link of Balon with stress conditions. At the least, discussion of this scenario (in the context of "future work") is warranted.

- A more specific title could help make the paper more recognizable, for example "An aeRF1-like family of bacterial ribosome hibernation factors". Or "...Pelota-like...". Manuscript title, of course, should be up to the authors.

Andrei Korostelev

Referee #3 (Remarks to the Author):

The authors have addressed all my previous concerns and I believe that this study is a nice contribution to a better understanding of ribosome hibernation.

Author Rebuttals to First Revision:

Below is our point-by-point response to the reviewers' comments.

Referees' comments:

Referee #1 (Remarks to the Author):

The authors have addressed all my previous concerns, improving dramatically the quality of the cryoEM structures as well as the characterization of the role of EF-Tu. They have additionally included new structures from human pathogenic bacteria, expanding the impact of this work. I consider the manuscript ready for publication in its current form.

Thank you very much for helping us to improve the manuscript.

Referee #2 (Remarks to the Author):

The revised manuscript describing the discovery of ribosome-binding stress factor Balon has substantially improved upon revision with the addition of new cryo-EM analyses, and biochemical and cellular assays; my previous concerns are well-addressed. I recommend publication of this insightful manuscript. I have just a few minor suggestions that should be straightforward for the authors to implement.

Thank you so much for your critical input!

Minor comments:

1. On page 12: "...causing a fifteen-fold reduction in the rate of protein synthesis (Fig. 2h)". Describe how the rates of protein synthesis were calculated.

We have now corrected this by adding the following description: ""...causing a fifteen-fold reduction in the rate of protein synthesis, as assessed by the relative levels of the reporter protein GFP (Fig. 2h)""

2. The short mention of Balon's overlap with antibiotic-binding sites and the corresponding panels 1g-l are confusing, in my opinion. The authors state that "...Balon's binding site overlaps with multiple ribosome-targeting antibiotics... (Fig. 1g-i). Thus, Balon ...occupies the ribosomal active sites, rendering them inaccessible to other molecules." It is not clear how antibiotics are relevant in the context of normal ribosome activity, at least as described in the text. By contrast, these figure panels may imply that (1) Balon is involved in antibiotic resistance; or (2) that antibiotics prevent Balon binding to the ribosome. Neither of these scenarios are tested in this work. To avoid confusion, I suggest removing this discussion and figure panels, or moving them to Supplementary information.

We agree. We have addressed this comment by deleting these panels (Fig. 1g-i) and by deleting their description in the text.

3. It is not clear from the density description which parts of the mRNA (in the P-tRNA-Balon structure) are resolved. Is the P-site codon resolved (despite heterogeneity, codon-anticodon base-pairing may have strong density; alternatively, this could be a specific codon – e.g. an AUG – which would suggest binding to initiating ribosomes)? How well is the mRNA codon/backbone resolved in the A site? mRNA tunnel? It would help readers if these findings are described clearly.

We agree. We have now added the following description of the density in the first section of the Results.

“The tRNA-bound ribosomes also contained density in the nascent peptide tunnel, indicating that these ribosomes are bound to the peptidyl-tRNA. In the mRNA channel, the density corresponding to mRNA could be traced along the A, P, and E sites, with the P-site signal showing a well-defined tRNA/mRNA base-pairing that corresponds more to heterogeneous rather than a specific type of tRNA/mRNA sequence. Overall, these data suggest that Balon can associate not only with vacant but also with elongating ribosomes.”

4. Discussion could include an important aspect of stress response in bacteria that may underlie the functional role of Balon. Specifically, cryo-EM/biochemical data here show that Balon binds EF-Tu in the presence of GDP – rather than in the presence of GTP. This suggests that Balon could be a sensor of the GDP/GTP ratio in the cell. Under normal growth conditions, the GDP-bound state of EF-Tu is rare due to quick exchange of GDP with more abundant GTP molecules. By contrast, the ratio of GDP/GTP increases during stress. Moreover, GTP concentration is reduced due to its conversion to pppGpp and subsequently to ppGpp. It may be physiologically relevant that the latter binds to GTPases similarly to GDP; therefore, the cellular substrate for Balon could be EF-Tu*ppGpp. Given the already impressive body of work in this manuscript, it is not required for the authors to do experiments to test whether Balon binds EF-Tu with ppGpp; but if they decide to do so, they might find the missing link of Balon with stress conditions. At the least, discussion of this scenario (in the context of “future work”) is warranted.

We have now included this idea in the final paragraph of our discussion. This paragraph is very brief because we were explicitly asked not to extend our text any further.

“Finally, the fact that EF-Tu only associates with Balon and hibernating ribosomes when it is bound to GDP suggests that the Balon/EF-Tu hibernation mechanism may represent a previously unknown mechanism for sensing intracellular levels of GTP, GDP, or possibly ppGpp to allow bacteria to initiate ribosome hibernation in response to an energy deficit.”

5. A more specific title could help make the paper more recognizable, for example “An aeRF1-like family of bacterial ribosome hibernation factors”. Or “...Pelota-like...”. Manuscript title, of course, should be up to the authors.

Thank you very much for this comment, we have left the title unchanged to make it more accessible for a general reader.

Referee #3 (Remarks to the Author):

The authors have addressed all my previous concerns and I believe that this study is a nice contribution to a better understanding of ribosome hibernation.

Thank you very much for your critical input and appreciation of our work!